# Thermodynamics of Evaporation from the Ocean Surface

**Rainer Feistel [1],\* and Olaf Hellmuth [2]**

[1] Leibniz Institute for Baltic Sea Research (IOW), Leibniz-Sozietät der Wissenschaften zu Berlin, 18119 Warnemünde, Germany
[2] Leibniz Institute for Tropospheric Research (TROPOS), 04318 Leipzig, Germany
\* Correspondence: rainer.feistel@io-warnemuende.de

**Abstract:** Adopted by the Intergovernmental Oceanographic Commission (IOC) of UNESCO in 2010 and the International Union of Geodesy and Geophysics (IUGG) in 2011, the Thermodynamic Equation of Seawater 2010 (TEOS-10) is the current geophysical standard for the thermodynamic properties of humid air, seawater and ice. TEOS-10 equations for evaporation and sublimation enthalpies are derived mathematically from the thermodynamic potential of a »sea air« model, denoting a multi-phase equilibrium composite of the geophysical aqueous mixtures. To estimating evaporation rates from the ocean, Dalton equations in various versions are implemented in numerical climate models. Some of those equations appear to be biased on climatic time scales if compared with proper thermodynamic driving forces. Such equations may lead to a spurious amplification of the hydrological cycle and an implied effect of cooling oceans. As an unbiased alternative, Dalton equations are proposed in terms of TEOS-10 relative fugacity (RF) or its conventional relative humidity (RH) approximations. With respect to RH uncertainties or trends, the substantial sensitivity of the evaporation flux may be estimated to be as much as 5 W m$^{-2}$ per 1 %rh. Within a maximum error of only 0.04 %rh, sea-surface RF may be approximated in terms of dew-point or frost-point temperatures using a simple formula.

**Keywords:** TEOS-10; Dalton equation; relative fugacity; climate model; hydrological cycle





## 1. Introduction

Evaporation from open water surfaces such as the oceans proceeds permanently and intensely but silently and invisibly. Most humans remain unaware of this process unless the water level is changing dramatically, as in the cases of the Aral and the Dead Sea. However, in 1687 Edmond Halley ([1] p. 368) revealed for the first time that "the whole Mediterranean must lose in Vapour, in a Summers-day, at least 5280 Millions of Tons". However, despite its fundamental importance, understanding marine evaporation still poses a severe challenge to climate research.

Owing to the complexity of physical processes in the atmosphere and hydrosphere, several numerical climate models possess uncertainties that exceed certain relevant, either observed or predicted, effects of global warming by orders of magnitude. Lauritzen et al. ([2] p. 2) found that some "models that conserve energy and water mass do not match an observed global mean precipitation rate. . . . The energy balance still suffers from significant errors. The causes of these errors are largely unknown, but are observed to be large over tropical oceans. . . . [In] pre-industrial simulations from a wide range of IPCC climate models, . . . most climate models featured biases of the order of 1 W m$^{-2}$ for the net global and the net atmospheric, oceanic, and land energy balances. . . . These imbalances are partly due to imperfect closure of the energy cycle in the fluid components." To demonstrate the significance of that uncertainty, a minor heating flux of just 0.005 W m$^{-2}$ is sufficient to raise the atmospheric temperature at an observed rate of 2 °C per century [3–7]. "The oceans have a heat capacity about 1000 times greater than the atmosphere and land surface" ([8] p. 420). If also expressed per global surface unit area, even an increase in the large oceanic

heat content by as much as 0.5 W m $^{-2}$ ([8] p. 427), [9,10] would remain well below the model uncertainty range. However, "the climate of the Earth is ultimately determined by the temperatures of the oceans" ([8] p. 420).

Of the water contained in the global troposphere, about 86% ([11] p. 60) or 85% [12] results from evaporation from the sea surface. While the total marine evaporation on the northern hemisphere is almost balanced against precipitation at sea, the precipitation over land is largely compensated by excess evaporation of the southern ocean ([11] p. 61). Zonally averaged, as estimated by Baumgartner and Reichel ([11] p. 80), from the sea, the strongest evaporation of 1318 mm yr$^{-1}$, or, equivalently, 104 W m$^{-2}$ of latent heat, occurs in the trade-wind belt between 15 and 25° N. An evaporation of 1540 mm yr$^{-1}$, or 122 W m$^{-2}$, occurs between 10 and 20° S, being particularly intense over the subtropical Indian Ocean.

"The sea surface interaction is obviously a highly significant quantity in simulating climate" ([13] p. 13). However, "the observational bases [of precipitation and evaporation] for the oceans are scanty and imprecise. . . . For evaporation there is a lack of certainty in the constants of the related formulas for calculations from climatological means or instantaneous meteorological observations." The uncertainty of at least 10 W m$^{-2}$ [14,15] with respect to energy fluxes across the air–sea interface is inadequate because "the by far largest part of heat is transferred to the air in the form of latent heat during subsequent condensation along with cloud formation. The heat budget over the sea is mainly controlled by the latent heat released to the air . . . The heat released to the air in latent form is larger by a multiple than the [sensible] heat transferred immediately to the air" (Original German text as quoted from Albrecht [16]: "Der weitaus größte Teil der Wärme wird der Luft in Form von latenter Wärme und nachfolgender Kondensation bei der Wolkenbildung zugeführt. . . . Der Wärmehaushalt der Luft über dem Meere wird . . . hauptsächlich durch die bei der Verdunstung an die Luft abgegebene latente Wärme bestimmt . . . Die an die Luft in latenter Form abgegebene Wärme ist dabei um ein Vielfaches größer als die durch den Austausch unmittelbar"). A typical evaporation of 1000 mm yr$^{-1}$ supplies the atmosphere (and cools down the ocean) with the latent water vapour heat at a rate of 79 W m$^{-2}$ per ocean surface area.

Observations and models of oceanic evaporation typically deviate from one another by 6 W m$^{-2}$, or 6% [17]. Reducing the systematic observational errors and the random uncertainties of ocean–atmosphere fluxes to less than 5 and 15 W m$^{-2}$ [15], respectively, is an ambitious target: "We need an accuracy of approximately ±15 W m$^{-2}$" ([18] p. 59).

Describing a complex natural evaporation process, the treatment of turbulent fluxes of momentum, sensible and latent heat, and of tracers in atmospheric models is part of the parameterisation of subgrid-scale processes. At the air–sea interface, these fluxes are usually specified as part of the boundary conditions and/or the surface layer parameterisation. In the majority of applications, the determination of fluxes across the air–sea interface relies on so-called bulk transfer formulations, which serve as a substitute for the downgradient ansatz or small-eddy approximation at the interface. In the bulk formulation, the turbulent flux of any quantity is parameterised as a product of two terms: (i) an aero- or thermodynamic driving force scaling with the local gradient of the quantity of interest, and (ii) an aerodynamic pre-factor describing the effectiveness of the turbulent flow to exchange the quantity of interest. The latter is a function of wind velocity considering the semi-empirical bulk-transfer coefficients. These transfer coefficients depend on the stability of the surface layer and can be parameterised using the Monin–Obukhov similarity theory (MOST). Under the assumption of horizontal flow homogeneity, a quasi-steady-state of turbulence, and altitude independence of turbulent momentum and heat fluxes in the near-surface layer, MOST describes turbulence in a thermally inhomogeneous medium by only four independent observables, namely the screening height, the friction velocity, the buoyancy parameter, and the sensible heat flux. A direct consequence of the MOST is the mutual interdependence of momentum, heat, and evaporation fluxes. In the ocean, the aerodynamic pre-factor depends on many determinants, such as meteorological factors and sea-surface roughness. The description of these complex dependencies is subject to

past and ongoing research. The subsequent analysis in this paper, however, focuses on the physical foundation of the thermodynamic driving force of the moisture flux. For the discussion of climate-change aspects, the aerodynamic pre-factor may be adjusted to the observed long-term global water balance, for which the assumption of adiabasis is safely justified.

This paper reviews selected thermodynamic aspects of ocean evaporation. Supporting the related studies and models, for the first time in the history of geophysics, internationally standardised physical key properties such as entropies, enthalpies and chemical potentials of humid air, seawater and ice have become quantitatively available from mutually consistent and accurate empirical thermodynamic potentials [19–24]. These formulations are jointly referred to as the »Thermodynamic Equation of Seawater—2010« (TEOS-10) [5,25–28], and are officially adopted and recommended by the International Association for the Properties of Water and Steam (IAPWS), the Intergovernmental Oceanographic Commission (IOC) of UNESCO, and the International Union of Geodesy and Geophysics (IUGG). In 2009 at Paris, IOC [29] particularly considered "the importance of an accurate formulation of the thermodynamics and equation of state of seawater as a fundamental component of ocean models, in particular for climate purposes". In 2011, at Melbourne, IUGG [30] urged "all marine scientists to use TEOS-10 . . . in their research and publications", considering "that since the International Thermodynamic Equation of Seawater—2010 (TEOS-10) has been adopted by the Intergovernmental Oceanographic Commission (IOC) at its 25th Assembly in June 2009 as the official description for the properties of seawater, of ice and of humid air". More recent improvements with respect to TEOS-10 are reviewed by Harvey et al. [31].

This paper is organised as follows. In Section 2, the typical Dalton equation used in recent numerical climate models to estimate evaporation fluxes is considered for the case of constant relative humidity (RH) at the marine surface. Under that condition, which is likely characteristic of the observed process of global warming, the Dalton coefficient is systematically rising along with the increase in temperature, numerically suggesting a putative acceleration of the hydrological cycle. In Section 3, the Dalton equation is stepwise derived from the fundamental equations of irreversible thermodynamics. The bias analysed in Section 2 seems to be introduced by a historical approximation that is suitable for short-term studies in the lab or surveys at sea but may prove problematic for long-term climate models. Rather than the usual humidity difference, the logarithm of RH is recommended as the driving force of an unbiased Dalton equation. Relative fugacity (RF) is an improved substitute for the climatological RH definition. In Section 4, the thermodynamic potentials of TEOS-10 for humid air, seawater and ice are combined in a multi-phase composite model to derive general thermodynamic expressions for the latent heat of evaporation and sublimation. Section 5 provides a simple version of the Dalton equation in terms of RF, making use of the latent heat equations of Section 4 in combination with the dew-point temperature of the sea-surface layer. In Appendices A–C, respectively, surface pressure Gibbs functions of the TEOS-10 equations of state for seawater, humid air and ice are quantitatively reported as empirical functions. Appendix D provides a list of the formula symbols used in this paper.

## 2. Dalton Equation: Climatological Bias?

From his laboratory experiments, John Dalton had concluded in 1798 that "the quantity of any liquid evaporated in the open air is directly as the force of steam from such liquid at its temperature, all other circumstances being the same" ([32] p. 537). This verbal law gave rise to what is presently known as the *Dalton equation* for evaporation [33–37]. In typical numerical climate models, a parameterisation of the upward latent heat flux density, $Q_L$, across the interface between humid air and a condensed aqueous phase (liquid water,

seawater or ice) takes the form of a modified Dalton equation, estimating the "force of steam" from the specific humidity of air [14,18,38–50], such as:

$$Q_L = LJ_W = L\rho^W Q_V = L\rho^W D(u) \left[ q_{eq}(S,T) - q \right] \tag{1}$$

Here, $T$ is the sea-surface temperature, $\rho^W$ is the mass density of liquid water (such as $\rho^W \approx 1000 \text{ kg m}^{-3}$), $L$ is the latent heat of evaporation (such as $L \approx (2500 - 2.5\, t/^\circ\text{C}) \text{ kJ kg}^{-1}$ for liquid water), $D$ is an empirical transfer coefficient (such as $D(u) = 1.2 \times 10^{-3} \times (\rho^{AV}/\rho^W) \times u$ ([18] p. 58)), $\rho^{AV}$ is the mass density of humid air, $u$ is the wind speed, $q_{eq}(S,T)$ is the specific (or absolute) humidity, which is the mass fraction of water vapour in humid air when in equilibrium with the condensed phase (such as a Clausius-Clapeyron formula [51]), and $q$ is the actual specific humidity of air above the ocean surface. The related mass-flux density of water evaporation is $J_W = Q_L/L$, expressed as mass per unit area and time, while the evaporation velocity (or evaporation rate), is $Q_V = J_W/\rho^W$; that is, the flux density of liquid water volume, typically measured in mm per day. Additionally, $S$ is the specific (or absolute) salinity, the mass fraction of dissolved salt in seawater, typically $S \approx 35 \text{ g kg}^{-1}$ in the oceans (note that, in TEOS-10, this »Absolute Salinity« is denoted by $S_A$ to clearly distinguish it from several alternative, mostly also unitless, oceanographic salinity scales [19]). In Section 4, expressions will theoretically be derived for the computation of the latent heat $L$ in the TEOS-10 framework. In Section 3, the relation between specific humidity and the theoretical thermodynamic driving force of evaporation will be analysed in more detail.

As compared to freshwater, salinity is lowering the vapour pressure of ocean water with salinities $S \approx 35 \text{ g kg}^{-1}$, according to Raoult's law [34,52], see also Sections 3 and 5 below:

$$q_{eq}(S,T) \approx 0.98\, q_{eq}(0,T) \tag{2}$$

The climatological definition of relative humidity [6,53,54] is

$$\psi_q = \frac{q}{q_{eq}(0,T)} \tag{3}$$

specified relative to the saturation humidity $q_{eq}(0,T)$, which occurs in equilibrium with either pure water or ice. Using this definition, the Dalton Equation (1) takes the form

$$Q_L = L\rho^W D(u) q_{eq}(0,T) \left(1 - \psi_q\right) \tag{4}$$

with respect to evaporating freshwater or sublimating ice, and

$$Q_L = L\rho^W D(u) q_{eq}(0,T) \left(0.98 - \psi_q\right) \tag{5}$$

with respect to evaporation from the ocean surface.

In agreement with observational experience, several climate models assume that the mean relative humidity at the ocean surface is constant at about $\psi_q \approx 80$ %rh, a value that is independent of global warming, season or latitude [8,55–58]. Then, in Equations (4) and (5), as a function of temperature, the value of $q_{eq}(0,T)$ is increasing at a rate of 7% K$^{-1}$, along with global warming, at climatological time scales, and this may substantially strengthen the predicted global evaporation. Consequently, assuming unchanged wind conditions, the use of Equations (4) and (5) in such a numerical climate model will simulate an amplification of oceanic evaporation as a direct consequence of globally rising temperatures. An intensified hydrological cycle is discussed in the climatological literature [7,39,48,58–64], but the implied putative cooling effect of stronger evaporation from the global ocean would be in contrast to measured data of ocean warming [9,10]. According to recent model comparison studies, "most CMIP6 [»Coupled Model Intercomparison Project Phase 6« [65]] models fail to provide as much heat into the ocean as observed" ([66] p. E1968). The simple mathematical example provided by Equations (4) and (5) may demonstrate how

easily minor changes or inappropriate approximations in the parameterisation of marine evaporation could result in systemically biased climate trend projections.

For illustration, one might refer to published long-term global-scale simulations of the latent heat flux (LHF), established on the basis of the Dalton equation, which give cause for serious concern. Multi-model-ensemble (MME) simulations comprising different state-of-the-art coupled general circulation models (CGCMs) revealed systematic differences in the LHF between model simulations and observations, without the possibility of drawing conclusive statements regarding the reason for these recognised biases [67]. For example, the global mean of LHF was found to be overestimated in the MME by 5.9 W m$^{-2}$, that is, 7.2% higher than the observations. Remarkable regional overestimations have been reported for latitudes from 10 to 20 °N, related to the excessive seasonal variations in LHF in the MME for the southwest branch of the Kuroshio Current, including the Kuroshio intrusion to the north of the South China Sea. Quite noticeable root–mean–square LHF errors appeared in coastal regions, such as at the west coast of Africa, the northwest coast of the Arabian Sea, the seas to the northeast of Japan, the equatorial eastern Pacific, and northeastern North America. Simulated long-term trends were analysed to be too weak compared to the observed ones, which were hypothesised to originate from uncertainties in both the thermodynamic driving force and the aerodynamic pre-factor, including wind velocity, bulk transfer coefficient, and the mass density of humid air. The most striking biases, however, were recognised for the specific humidity, $q$, and the wind velocity, $u$; the simulated rate of the increase in $q$ was a factor of six larger than the observed one, while the simulated rate of increase in $u$ was only half the observed one. Both biases in these trends tend to underestimate the LHF. The reasons for the poor MME performance in the simulations of $q$ and $u$, again, remain hidden; Zhang et al. [67] concluded that "accordingly, additional exploration is required to enhance our knowledge of the biases in $q$ and $u$ and to find ways to improve this problem in the models as much as possible." This includes enhanced attention to the spread in observational data, especially surface humidity, one of the most important factors influencing the biases in LHF.

In the following Section, the Dalton Equation (1) will be derived from fundamental equations of irreversible thermodynamics, in combination with TEOS-10. Subsequent mathematical approximation steps are critically discussed, leading to the conclusion that even if the common Dalton equation is an appropriate tool for short-time lab experiments and field observations, care should be taken before implementing it incautiously in numerical climate models for long-term predictions.

## 3. Evaporation: Thermodynamic Driving Force and Approximations

Using the framework of linear irreversible thermodynamics, fluxes in matter and heat, $J_k$, are described by linear combinations of so-called Onsager forces, $X_k$, in the form

$$J_k = \sum \Omega_{kl} X_l \tag{6}$$

Except for the rotating reference frames and magnetic fields, the matrix of Onsager coefficients is always symmetric, $\Omega_{kl} = \Omega_{lk}$, and positive definite, so that the local entropy production, measured in W K$^{-1}$ m$^{-3}$ in the case of basic SI units,

$$\sigma = \sum J_k X_k = \sum \Omega_{kl} X_k X_l \geq 0 \tag{7}$$

is positive except at equilibrium, when all the forces vanish ([68] Chapter IV). According to Prigogine's theorem ([69] Section 9.3), $\sigma$ takes a minimum value at steady states if the linearity (6) holds.

In this Section, Equation (6) will be compared with some versions of the Dalton equation that are in practical use for estimating the evaporation rate from the ocean surface, and the related Onsager coefficient will be derived from the related empirical transfer coefficients. Although textbooks on non-equilibrium thermodynamics usually restrict this formalism to irreversible processes occurring within single continuous phases, here, the

theory is assumed to also be applicable to fluxes across phase boundaries. The natural relative humidity of about 80 %rh above the ocean surface is sufficiently close to saturation, and the typical air–sea temperature difference is mostly low [70], so that, for the upward fluxes, $J_W$ and $J_Q$, in water-vapour mass and heat, respectively, across the air–interface, the linear Onsager regime may be assumed to be a reasonable *first approximation*. The associated forces in $z$-direction are ([69] Equation 2.21) ([71] Equation 22.49),

$$X_W = -\frac{\partial}{\partial z}\left(\frac{\mu_W}{T}\right), \text{ and} \tag{8}$$

$$X_Q = \frac{\partial}{\partial z}\left(\frac{1}{T}\right) \tag{9}$$

Here, $\mu_W$ is the chemical potential of water. Assuming, as a *second approximation*, a two-box model of humid air (indexed by AV) in an upper box and seawater (by SW) in the lower one, the forces are

$$X_W = -\frac{1}{\lambda}\left[\frac{\mu_W^{AV}(A, T_{AV})}{T_{AV}} - \frac{\mu_W^{SW}(S, T_{SW})}{T_{SW}}\right], \text{ and} \tag{10}$$

$$X_Q = \frac{1}{\lambda}\left(\frac{1}{T_{AV}} - \frac{1}{T_{SW}}\right) \tag{11}$$

Here and below, the pressure dependence of functions is mostly omitted for simplicity because all equations refer to the atmospheric pressure of the standard ocean, $p = p_{SO}$, at its surface. The mass fraction of dry air in humid air is $A$; that of dissolved salt in seawater is $S$. Further, $\lambda$ is the thickness of a fictitious membrane separating the boxes, which is only penetrable by water molecules. Variables $T_{AV}$ and $\mu_W^{AV}$, respectively, denote the temperature and chemical potential of water per unit mass in the humid-air box, and $T_{SW}$ and $\mu_W^{SW}$ are the properties of the seawater box. Both boxes are assumed to have the same pressure, and each are presumed to be homogeneously filled due to turbulent mixing, so that the mutual exchange rates, $J_W$ and $J_Q$, are controlled by interface properties rather than by any transport processes inside the box volumes.

The intention behind this model is to derive approximation formulas for the climatological, globally averaged evaporation rate. The air–sea temperature difference is mostly small (mostly within 2 K) and has a varying sign [70]. Let a *third approximation level* consist of isothermal conditions, $T_{AV} \approx T_{SW} = T$, implying a vanishing sensible heat exchange, $X_Q = J_Q = 0$. Note that certain numerical models only apply the isothermal condition to equations for the evaporation flux, but still (and inconsistently) permit sensible heat fluxes driven by air–sea temperature offsets.

Approximate isothermal evaporation occurs with the mass-flux density [72],

$$J_W = -\frac{\Omega_{WW}}{\lambda T}\left[\mu_W^{AV}(A, T) - \mu_W^{SW}(S, T)\right] \tag{12}$$

Consistent with Equation (10), $X_W$ is measured in J kg$^{-1}$ K$^{-1}$ m$^{-1}$ if expressed in basic SI units and consequently, $J_W$ in kg m$^{-2}$ s$^{-1}$, according to Equation (7).

The difference in Equation (12) is also known as the *affinity of vapourisation* ([72] p. 42). TEOS-10 provides quantitative values for both chemical potentials, but the empirical transfer coefficient $\Omega_{WW}/\lambda$ still needs to be determined from observations. For this purpose, the chemical potentials may be exactly expressed in terms of the fugacities $f_V^{AV}$ of water vapour in humid air, and $f_W^{SW}$ of liquid water in seawater [73,74]

$$\mu_W^{AV}(A, T, p) = \mu_W^{id}(T, x_V p) + R_W T \ln\frac{f_V^{AV}(x_V, T)}{x_V p}, \text{ and similarly,} \tag{13}$$

$$\mu_W^{SW}(S, T, p) = \mu_W^{id}(T, x_W p) + R_W T \ln \frac{f_W^{SW}(x_W, T)}{x_W p} \tag{14}$$

Fugacity, as introduced by Lewis [75], is the real-gas equivalent of the ideal-gas partial pressure [76]. The specific gas constant of water is $R_W = R/M_W$ in terms of the molar gas constant $R$ and the molar mass $M_W$ of water. The mole fraction of water vapour in humid air is,

$$x_V(A) = \left[1 + \frac{A M_W}{(1-A) M_A}\right]^{-1} \tag{15}$$

in which $M_A$ is the molar mass of dry air. Similarly,

$$x_W(S) = \left[1 + \frac{S M_W}{(1-S) M_S}\right]^{-1} \tag{16}$$

is the mole fraction of water in seawater in which the molar mass of sea salt is $M_S$. The ideal gas chemical potential of pure water, $\mu_W^{id}(T, p)$, is defined by the asymptotic low-pressure limit [76],

$$\mu_W^{id}(T, p) = R_W T \ln \frac{p}{1\,\text{Pa}} + \lim_{p \to 0}\left\{\mu_W(T, p) - R_W T \ln \frac{p}{1\,\text{Pa}}\right\} \tag{17}$$

so that

$$\mu_W^{id}(T, x_V p) - \mu_W^{id}(T, x_W p) = R_W T \ln \frac{x_V}{x_W} \tag{18}$$

An explicit formula for $\mu_W^{id}(T, x_V p)$ in the context of TEOS-10 is available from Equation (A35) in Appendix B. Using Equations (13), (14) and (18), the evaporation flux from Equation (12) in terms of fugacities takes the form

$$J_W = -D_f \ln \frac{f_V^{AV}(x_V, T)}{f_W^{SW}(x_W, T)} \tag{19}$$

The fugacity-based mass transfer coefficient is $D_f = R_W \Omega_{WW}/\lambda$. For comparison with common empirical evaporation estimates, certain approximations of the thermodynamically correct Equation (19) are required for the two-box model.

The *fourth approximation level* exploits the Lewis Rule [73] for fugacities of dilute, approximately ideal solutions, $S \ll 1$. Using Equation (16), the rule states that

$$f_W^{SW}(x_W, T) \approx x_W(S)\, f_W^{SW}(1, T) \approx \left[1 + \frac{M_W}{M_S} S + O\!\left(S^2\right)\right]^{-1} f_W^{SW}(1, T) \tag{20}$$

The fugacity of pure water, $f_W^{SW}(1, T)$, equals the fugacity of water vapour in saturated humid air,

$$f_W^{SW}(1, T) = f_V^{AV}(x_V^{sat}, T) \tag{21}$$

In this way, the evaporation Equation (19) is expressed as

$$J_W \approx -D_f \ln\left[\left(1 + \frac{M_W}{M_S} S\right) \psi_f(x_V, T)\right] = -D_f\left[\ln\left(1 + \frac{M_W}{M_S} S\right) + \ln \psi_f(x_V, T)\right] \tag{22}$$

Here, as an activity-based definition of relative humidity (RH), the *relative fugacity* (RF) is introduced [77],

$$\psi_f(x_V, T) \equiv \frac{f_V^{AV}(x_V, T)}{f_V^{AV}(x_V^{sat}, T)} \tag{23}$$

A rough estimate of the value of the transfer coefficient, $D_f$, can be obtained from Equation (22) assuming a global mean climatological evaporation rate of $Q_V = 1200\,\text{mm yr}^{-1}$,

an ocean salinity of $S = 35$ g kg$^{-1}$ and a marine surface relative humidity of $\psi_f = 80$ %rh, to give

$$D_f = \frac{\Omega_{WW}}{\lambda} R_W = -\frac{Q_V \rho^W}{0.02 + \ln 0.8} = 5.91 \text{ m yr}^{-1} \rho^W = 187 \times 10^{-9} \text{ m s}^{-1} \rho^W \quad (24)$$

These observations have supported the assumption that the climatological values of $J_W$, $S$, and $\psi_f$ are largely unaffected by the global warming trend. Consequently, this may also be concluded for the value of $D_f$.

A subsequent *fifth approximation step* is based on the assumption of nearly saturated humid air, that is, $1 - \psi_f \ll 1$, so

$$\ln \psi_f \approx \psi_f - 1 \quad (25)$$

and, accordingly, the evaporation flux (22) may be estimated by virtue of $\ln\left(1 + \frac{M_W}{M_S} S\right) \approx \frac{M_W}{M_S} S$

$$J_W \approx D_f \left[1 - \frac{M_W}{M_S} S - \psi_f\right] \quad (26)$$

In a *sixth approximation step*, RF is replaced by the conventional metrological and meteorological RH [6],

$$\psi_f \approx \psi_x \equiv \frac{x_V}{x_V^{sat}} = \frac{e}{e^{sat}} \quad (27)$$

Here, $e = x_V p$ is the partial pressure of water vapour in humid air. The difference $\left|\psi_f - \psi_x\right|$ is caused by deviations from ideal gas properties and is mostly within the uncertainty found in typical RH measurements [6].

Under the assumption of $\psi_f \approx \psi_x$, Equation (26) takes the form of the Dalton [32] equation,

$$J_W \approx D_e \left[\left(1 - \frac{M_W}{M_S} S\right) e^{sat}(T) - e\right] \quad (28)$$

Many later authors used this type of equation with different expressions for the vapour-pressure-based transfer coefficient,

$$D_e = \frac{D_f}{e^{sat}(T)} = \frac{\Omega_{WW}}{\lambda} \frac{R_W}{e^{sat}(T)} \quad (29)$$

such as Trabert [33], Sverdrup [78,79], Albrecht [80], Budyko [81], Debski [35], Foken et al. [82], Foken [83], Littmann et al. [37] or Bernhofer et al. [84]. For example, as linear functions of the wind speed, $u$, Jacobs [85] suggested, for the northern oceans [11],

$$D_e/\rho^W = 0.143 \frac{\text{mm day}^{-1}}{\text{mbar}} \times \frac{u}{\text{m s}^{-1}} = 16.55 \times 10^{-12} \text{ Pa}^{-1} \times u \quad (30)$$

while, for the southern oceans, Privett [86,87] proposed,

$$D_e/\rho^W = 0.00587 \frac{\text{cm day}^{-1}}{\text{mbar}} \times \frac{u}{\text{kn}} = 13.2 \times 10^{-12} \text{ Pa}^{-1} \times u \quad (31)$$

For a typical saturation pressure about $e^{sat} = 1$ kPa and a wind speed of $u = 10$ m s$^{-1}$, the related values of $D_f = D_e e^{sat}$, respectively, from Equation (29) are $D_f/\rho^W = 165.5 \times 10^{-9}$ m s$^{-1}$ and $D_f/\rho^W = 132 \times 10^{-9}$ m s$^{-1}$, which roughly agree with the guessed global mean value, Equation (24).

More important than those offsets, however, appears to be that, physically, the value of $D_f$, Equation (24), is likely free of systematic trends resulting from global warming. If, by contrast, a climate model implements the Dalton Equation (28), with $D_e$ being independent

of the climatic temperature rise, such as in Equations (30) or (31), the physically significant $D_f = D_e \, e^{\mathrm{sat}}(T)$ will increase by about 7% per kelvin along with global warming. *In a climate model, implementation of the traditional but approximate Dalton Equation (28) may lead to a mathematically pretended, spurious ocean cooling as a result of the systematically intensifying evaporation rates controlled by the physically more justified Equation (26).*

As an alternative *sixth approximation step*, RF is replaced by the conventional climatological RH [6,53,54],

$$\psi_f \approx \psi_q \equiv \frac{q}{q^{\mathrm{sat}}} \tag{32}$$

Here, $q = 1 - A$ is the specific humidity. Note that the difference $\left|\psi_f - \psi_q\right|$ is systematic and may even exceed the uncertainty of typical RH [6].

With the approximation $\psi_f \approx \psi_q$, Equation (26) takes the form,

$$J_\mathrm{W} \approx D_q \left[ \left(1 - \frac{M_\mathrm{W}}{M_\mathrm{S}} S \right) q^{\mathrm{sat}}(T) - q \right] \tag{33}$$

More recently, particularly in numerical models, several authors [18,39–41,43,45,50] use this type of equation with varying expressions for the humidity-based transfer coefficient,

$$D_q = \frac{D_f}{q^{\mathrm{sat}}(T)} = \frac{\Omega_\mathrm{WW}}{\lambda} \frac{R_\mathrm{W}}{q^{\mathrm{sat}}(T)} \tag{34}$$

For example, at 25 °C, the marine RH of 80 %rh corresponds to $q^{\mathrm{sat}} - q \approx 0.4\%$. Numerical values for the coefficients are given by Stewart ([18] p. 58) and Smith [40] in the form of

$$D_q = \rho^{\mathrm{AV}} C_L u \tag{35}$$

involving the latent heat transfer coefficient

$$C_L = 1.2 \times 10^{-3} \tag{36}$$

As above, the fact that, physically, the value of $D_f$, Equation (24), is likely free of systematic trends resulting from global warming raises concerns about the climatological use of constants such as (35) or (36). If, by contrast, a climate model implements the Dalton Equation (33) with $D_q$ being independent of the climatic temperature rise, the physically significant $D_f = D_q \, q^{\mathrm{sat}}(T)$ will increase by about 7% per kelvin along with global warming. *In a climate model, implementation of the traditional but approximate Dalton Equation (33) may lead to a mathematically pretended, spurious ocean cooling as a result of the systematically intensifying evaporation rates controlled by the physically more justified Equation (26).*

Dalton equations of the form (28) or (33) have frequently and successfully been used in the previous century to estimate short fluxes in in vitro experiments or in situ observations. In such cases, the deviation of $D_f$ from $D_e$ or $D_q$ is just a constant factor. In implementations of climate models, however, this factor may no longer be assumed to be constant and will imply a systematic bias between the corresponding model predictions for trends in ocean evaporation and the global hydrological cycle.

To avoid spurious trends in evaporation being silently induced by short-term Dalton equations being embedded in long-term climate models without precaution, we suggest implementing these equations in the thermodynamic form of Equation (12)

$$J_\mathrm{W} = -\frac{D_f}{R_\mathrm{W} T} \left[ \mu_\mathrm{W}^{\mathrm{AV}}(A, T) - \mu_\mathrm{W}^{\mathrm{SW}}(S, T) \right] \tag{37}$$

if TEOS-10 chemical potentials are numerically available [88,89], or in the form of Equation (22),

$$J_\mathrm{W} = -D_f \left[ \frac{M_\mathrm{W}}{M_\mathrm{S}} S + \ln \psi_f(x_\mathrm{V}, T) \right] \tag{38}$$

if the code for the relative fugacity of TEOS-10 is available [90]. Alternatively, Equation (69) could be used, or $\psi_x$ or $\psi_q$ could be used as substitutes of $\psi_f$ therein, such as

$$J_{\mathrm{W}} \approx -D_f \left( \frac{M_{\mathrm{W}}}{M_{\mathrm{S}}} S + \ln \frac{q}{q^{\mathrm{sat}}} \right) \tag{39}$$

The transfer coefficient $D_f$ may be estimated from available formulas for $D_e$ or $D_q$, respectively, at time-independent, present-day reference temperatures $T_{\mathrm{ref}}$ by $D_f = D_e\, e^{\mathrm{sat}}(T_{\mathrm{ref}})$ or $D_f = D_q\, q^{\mathrm{sat}}(T_{\mathrm{ref}})$.

Equations similar to Equation (38) in terms of using RH as the driving force for evaporation, rather than the vapour–pressure difference, have frequently been suggested, for example, by Romanenko [91], Littmann et al. [37], Oudin et al. [92], Feistel and Ebeling [93], Bernhofer et al. [84] or Feistel and Hellmuth [7].

## 4. Climatological Hydrosphere: Thermodynamics of »Sea Air«

### 4.1. Equation of State of »Sea Air«

The physical processes of weather and climate occur between the solid crust of the Earth and the cosmic space beyond the top of the atmosphere. Using reasonable approximations, this volume contains three phases, solid, liquid and gaseous, and three substances, water, dry air and sea salt. Each of those were considered to have a fixed chemical and isotopic composition. Thermodynamically, this system may be regarded as the »climatological hydrosphere«, or »sea-air system« for short [24], including oceans and atmosphere, surface waters, ice cover, and clouds. Due to the energy and entropy exchanges that occur across its boundaries, this is an open, non-equilibrium system under gravity but has a negligible gain in or loss of matter. However, sufficiently small subsystems can be successfully described as local thermodynamic equilibria, as is achieved by TEOS-10.

J. Willard Gibbs [94,95] discovered that all thermodynamic equilibrium properties of such a »sea-air system« can, at least in principle, be derived from a single mathematical function, namely, by one of its so-called »thermodynamic potentials«. In the framework of the canonical ensemble of statistical thermodynamics, this single multi-phase potential function is the Helmholtz energy of sea air,

$$F^{\mathrm{SA}}\left(T, V, m^{\mathrm{H_2O}}, m^{\mathrm{A}}, m^{\mathrm{S}}\right) = F^{\mathrm{SA,id}}\left(T, V, m^{\mathrm{H_2O}}, m^{\mathrm{A}}, m^{\mathrm{S}}\right) - k_{\mathrm{B}} T \ln Q^{\mathrm{SA}} \tag{40}$$

The system with the masses $m^{\mathrm{H_2O}}, m^{\mathrm{A}}, m^{\mathrm{S}}$, respectively, of water, dry air and sea salt contained in a volume $V$ may be at thermodynamic equilibrium at the temperature $T$. The function $F^{\mathrm{SA,id}}$ is the Helmholtz energy calculated from well-known equations for ideal gas mixtures [96,97]. The Boltzmann constant, $k_{\mathrm{B}}$, is exactly and ultimately defined in the latest SI [98] as the conversion factor between the energy units of kelvin and joule. To obtain the canonical partition function of Equation (40), the $6N$-dimensional configuration integral

$$Q^{\mathrm{SA}}\left(N^{\mathrm{H_2O}}, N^{\mathrm{A}}, N^{\mathrm{S}}, T, V\right) = \frac{1}{(4\pi V)^N} \int e^{-U(q)/(k_{\mathrm{B}}T)}\, \mathrm{d}q \tag{41}$$

should be carried out over all possible positions and angular orientations $q$ of the $N = N^{\mathrm{H_2O}} + N^{\mathrm{A}} + N^{\mathrm{S}}$ particles of water, air and salt, respectively, inside the volume $V$, evaluating the potential energy $U(q)$ of each particular spatial arrangement of the molecules. These configurations include homogeneous states as well as states consisting of separate phases, of which the thermodynamically stable states provide the dominating contributions to the integral (41). For this reason, from the potential function $F^{\mathrm{SA}}$, the latent heats of these phase transitions can be mathematically derived by varying the temperature $T$ and analysing the related mass transfer from one phase to another.

The integral $Q^{\mathrm{SA}}$ given by Equation (41) is a unique function of its arguments, which mean that a single thermodynamic potential function is sufficient to mathematically describe all thermodynamic equilibrium properties of the multi-phase composite mixture "sea

air". In practice, the potential energy $U(\mathbf{q})$ of water, salt and air is not known sufficiently well, nor is the execution of the high-dimensional configuration integral possible. Rather, $F^{SA}$ is empirically estimated from various thermodynamic measurements. In the reasonable TEOS-10 approximation, it is additionally assumed that dry air is only admixed with water vapour and sea salt is only admixed with liquid water, so that ice, if present, consists solely of pure water. Then,

$$F^{SA} = \left(m^S + m^W\right) f^{SW}\left(S, T, \rho^{SW}\right) + \left(m^A + m^V\right) f^{AV}\left(A, T, \rho^{AV}\right) + m^{Ih} f^{Ih}\left(T, \rho^{Ih}\right) \quad (42)$$

is the Helmholtz energy of a composite equilibrium system, with the optional inclusion of volume portions of humid air, seawater and ambient hexagonal ice Ih. The partial masses of liquid, $m^W$, gaseous, $m^V$, and solid water, $m^{Ih}$, as well as the associated partial volumes $V^{SW}$, $V^{AV}$ and $V^{Ih}$ occupied by the three different phases, are functions of the given properties $\left(T, V, m^{H_2O}, m^A, m^S\right)$. This means that the mutual phase-equilibrium conditions are obeyed. Depending on $T$ and $p$, the ice phase may or may not be found; liquid may be absent only if $m^S = 0$, and vapour needs to exist unless $m^A = 0$. Salinity, i.e., the mass fraction of dissolved salt in seawater, is $S = m^S / \left(m^S + m^W\right)$. The mass fraction of dry air in humid air is $A = m^A / \left(m^A + m^V\right)$. The partial densities, $\rho^{SW} = \left(m^S + m^W\right) / V^{SW}$, $\rho^{AV} = \left(m^A + m^V\right) / V^{AV}$, and $\rho^{Ih} = m^{Ih} / V^{Ih}$ serve as independent variables of the phase-specific Helmholtz functions $f^{SW}$, $f^{AV}$, $f^{Ih}$ of the phases.

If gravity is ignored for sufficiently small parcels, the equal pressure, $p = \rho^2 (\partial f / \partial \rho)_T$, of all phases, each possessing a particular mass density $\rho$ and a Helmholtz function $f$, is one of the equilibrium conditions. Then, it is convenient to use the Gibbs functions as the thermodynamic potential, as defined by the Legendre transform $g(T, p) = f(T, \rho) + p / \rho$, and replacing the independent variable $\rho$ with $p$. The related Gibbs energy of sea air

$$G^{SA}\left(T, p, m^{H_2O}, m^A, m^S\right) = F^{SA}\left(T, V, m^{H_2O}, m^A, m^S\right) + pV, \quad (43)$$

is additive

$$G^{SA} = G^{SW} + G^{AV} + G^{Ih} \quad (44)$$

with respect to the separate liquid, gaseous and solid portions,

$$G^{SW}\left(m^S, m^W, T, p\right) = \left(m^S + m^W\right) g^{SW}(S, T, p) \quad (45)$$

$$G^{AV}\left(m^A, m^V, T, p\right) = \left(m^A + m^V\right) g^{AV}(A, T, p) \quad (46)$$

$$G^{Ih}\left(m^{Ih}, T, p\right) = m^{Ih} g^{Ih}(T, p) \quad (47)$$

TEOS-10 provides explicit empirical equations for the Gibbs functions $g^{SW}(S, T, p)$ and $g^{Ih}(T, p)$. The function $g^{AV}(A, T, p)$ is available as a virial approximation derived from the TEOS-10 equation for $f^{AV}(A, T, \rho)$; see Appendix B.

The exchange of water between the different phases vanishes at equilibrium if their chemical potentials of water

$$\mu_W^{SW} = \left(\frac{\partial G^{SW}}{\partial m^W}\right)_{m^S, T, p} = g^{SW}(S, T, p) - S\left(\frac{\partial g^{SW}}{\partial S}\right)_{T, p} \quad (48)$$

$$\mu_V^{AV} = \left(\frac{\partial G^{AV}}{\partial m^V}\right)_{m^A, T, p} = g^{AV}(A, T, p) - A\left(\frac{\partial g^{AV}}{\partial A}\right)_{T, p} \quad (49)$$

$$\mu^{Ih} = \left(\frac{\partial G^{Ih}}{\partial m^{Ih}}\right)_{T, p} = g^{Ih}(T, p) \quad (50)$$

take the same values, i.e.,

$$\mu_{\mathrm{W}}^{\mathrm{SW}}(S, T, p) = \mu_{\mathrm{V}}^{\mathrm{AV}}(A, T, p) = \mu^{\mathrm{Ih}}(T, p) \tag{51}$$

These equations determine the equilibrium values of $S$ and $A$ at given $T$ and $p$, along with the conservation of the total water mass,

$$m^{\mathrm{H_2O}} = m^{\mathrm{W}} + m^{\mathrm{V}} + m^{\mathrm{Ih}} = const \tag{52}$$

These conditions are simplified if certain sea–air phases do not exist, such as missing ice at higher temperatures, or missing liquid water in the cryosphere.

While the potentials $F^{\mathrm{SA}}$, as well as $G^{\mathrm{SA}}$, are, in principle, fully determined within statistical thermodynamics, the related empirical functions include six free constants, resulting from the fact that only differences in energies and entropies, rather than their absolute values, can be derived from the measurement of each included substance [99]. As the values of those constants do not affect the prediction of any measurable properties, they can be conveniently specified by arbitrary reference-state conditions. In TEOS-10, these conditions are given by the vanishing internal energy and entropy of liquid water at the triple point, as well as the vanishing enthalpy and entropy of seawater and dry air at the standard ocean state [5,22,23,88,100,101]. To ensure the correctness of $F^{\mathrm{SA}}$ and $G^{\mathrm{SA}}$, these conditions must hold identically for each of the three substances, regardless of the phase in which they are contained.

However, beyond TEOS-10, the chemical or isotopic compositions of water, dry air, or sea salt may significantly change during geophysical processes. If this occurs, suitable reference states must be separately specified for the groups of species as involved in those changes. This may occur, for example, if the air dissolved in water needs to be considered, or precipitating salts in concentrated brines, or if the Standard Light Antarctic Precipitation Water (SLAP) must be isotopically distinguished from Standard Mean Ocean Water (SMOW). The composition of sea salt is rather constant in the ocean; however, significant deviations from the *Reference Composition* [102] are caused by dissolved silicate in the deep Pacific [103,104], or lime in the Baltic Sea [105]. Metrologically, the effect of the rising concentration of carbon dioxide on the density of moist air is accounted for by Picard et al. [106].

### 4.2. Latent Heats of Phase Transitions

At a constant surface pressure, the key quantity governing the energy balance is the enthalpy of sea air,

$$H^{\mathrm{SA}} \equiv G^{\mathrm{SA}} - T\left(\partial G^{\mathrm{SA}}/\partial T\right)_{p, m^{\mathrm{H_2O}}, m^{\mathrm{A}}, m^{\mathrm{S}}} = H^{\mathrm{SW}} + H^{\mathrm{AV}} + H^{\mathrm{Ih}} \tag{53}$$

The enthalpies of the different phases are, in terms of the TEOS-10 Gibbs functions,

$$H^{\mathrm{SW}} = \left(m^{\mathrm{S}} + m^{\mathrm{W}}\right) h^{\mathrm{SW}}(S, T, p) \equiv \left(m^{\mathrm{S}} + m^{\mathrm{W}}\right)\left[g^{\mathrm{SW}} - T\left(\frac{\partial g^{\mathrm{SW}}}{\partial T}\right)_{S,p}\right] \tag{54}$$

$$H^{\mathrm{AV}} = \left(m^{\mathrm{A}} + m^{\mathrm{V}}\right) h^{\mathrm{AV}}(A, T, p) \equiv \left(m^{\mathrm{A}} + m^{\mathrm{V}}\right)\left[g^{\mathrm{AV}} - T\left(\frac{\partial g^{\mathrm{AV}}}{\partial T}\right)_{A,p}\right] \tag{55}$$

$$H^{\mathrm{Ih}} = m^{\mathrm{Ih}} h^{\mathrm{Ih}}(T, p) \equiv m^{\mathrm{Ih}}\left[g^{\mathrm{Ih}} - T\left(\frac{\partial g^{\mathrm{Ih}}}{\partial T}\right)_{p}\right] \tag{56}$$

The change of enthalpy with temperature at constant pressure is the isobaric heat capacity. To compute this property of sea air, the transfer of water between the phases in

dependence on the temperature needs to be allowed for. The temperature derivative of $H^{\mathrm{SA}}$ results in

$$
\begin{aligned}
C_p^{\mathrm{SA}} \equiv \left(\frac{\partial H^{\mathrm{SA}}}{\partial T}\right)_{p,m^{\mathrm{H_2O}},m^{\mathrm{A}},m^{\mathrm{S}}} &= \left(m^{\mathrm{S}}+m^{\mathrm{W}}\right)c_p^{\mathrm{SW}} + \left(m^{\mathrm{A}}+m^{\mathrm{V}}\right)c_p^{\mathrm{AV}} + m^{\mathrm{Ih}}c_p^{\mathrm{Ih}} \\
&- \left(\frac{\partial m^{\mathrm{W}}}{\partial T}\right)_{p,m^{\mathrm{H_2O}},m^{\mathrm{A}},m^{\mathrm{S}}} L^{\mathrm{evap}} - \left(\frac{\partial m^{\mathrm{Ih}}}{\partial T}\right)_{p,m^{\mathrm{H_2O}},m^{\mathrm{A}},m^{\mathrm{S}}} L^{\mathrm{subl}}
\end{aligned}
\tag{57}
$$

Here, $c_p^{\mathrm{SW}} \equiv \left(\partial h^{\mathrm{SW}}/\partial T\right)_{S,p}$, $c_p^{\mathrm{AV}} \equiv \left(\partial h^{\mathrm{AV}}/\partial T\right)_{A,p}$ and $c_p^{\mathrm{Ih}} \equiv \left(\partial h^{\mathrm{Ih}}/\partial T\right)_p$, respectively, are the isobaric specific heat capacities of seawater, humid air, and ice Ih. Exploiting the conservation of the total water mass, Equation (52),

$$
\frac{\partial m^{\mathrm{H_2O}}}{\partial T} = 0 = \frac{\partial m^{\mathrm{W}}}{\partial T} + \frac{\partial m^{\mathrm{V}}}{\partial T} + \frac{\partial m^{\mathrm{Ih}}}{\partial T}
\tag{58}
$$

the latent heats in Equation (57), i.e., the isobaric specific evaporation and sublimation enthalpies, respectively, follow to be given by the expressions [24]

$$
L^{\mathrm{evap}} \equiv h^{\mathrm{AV}} - A\left(\frac{\partial h^{\mathrm{AV}}}{\partial A}\right)_{T,p} - h^{\mathrm{SW}} + S\left(\frac{\partial h^{\mathrm{SW}}}{\partial S}\right)_{T,p}
\tag{59}
$$

and

$$
L^{\mathrm{subl}} \equiv h^{\mathrm{AV}} - A\left(\frac{\partial h^{\mathrm{AV}}}{\partial A}\right)_{T,p} - h^{\mathrm{Ih}}
\tag{60}
$$

If necessary, the rates of evaporation, $\partial m^{\mathrm{W}}/\partial T$, and of sublimation, $\partial m^{\mathrm{Ih}}/\partial T$, appearing in Equation (57) are available from the temperature derivative of the equilibrium condition, Equation (51).

If humid air is approximated by an ideal mixture, i.e.,

$$
h^{\mathrm{AV}} \approx h^{\mathrm{AV,\,id}}(A,T,p) = Ah^{\mathrm{A}}(T,p) + (1-A)h^{\mathrm{V}}(T,p)
\tag{61}
$$

the familiar simple expression of vapour's contribution to the latent heat can be derived as follows:

$$
h^{\mathrm{AV,\,id}} - A\left(\frac{\partial h^{\mathrm{AV,\,id}}}{\partial A}\right)_{T,p} = h^{\mathrm{V}}(T,p)
\tag{62}
$$

even if $h^{\mathrm{A}}$ or $h^{\mathrm{V}}$ themselves are not given in ideal-gas approximations.

Numerous additional details of the thermodynamic properties related to the mutual phase transitions between seawater, ice and humid air are outlined by Feistel et al. [5,24,88]. For the computation of those properties, an open-source code is available from the TEOS-10 SIA Library [25,89].

## 5. Relative Fugacity Approximation

The Dalton equation in the form of Equation (22) permits estimation of the climate sensitivity with respect to sea-surface RH. Assuming that, approximately, $D_f \approx 200 \times 10^{-9}\,\mathrm{m\,s^{-1}} \times \rho^{\mathrm{W}}$ from Equation (24), the related latent heat flux fluctuation,

$$
|\delta Q_L| = L\,|\delta J_{\mathrm{W}}| = LD_f\left|\frac{\delta\psi_f}{\psi_f}\right| \approx 500\,\mathrm{W\,m^{-2}}\left|\frac{\delta\psi_f}{\psi_f}\right|
\tag{63}
$$

indicates that an error of 1% in RH, which is about the meteorological measurement uncertainty, would cause an error of 5 W m$^{-2}$ in the computed ocean atmosphere latent heat fluxes. For comparison, the observed global warming of the atmosphere is driven by a minor climatic forcing of only 0.005 W m$^{-2}$, the total anthropogenic power consumption amounts to 0.02 W m$^{-2}$, and the ocean is warming up by 0.5 W m$^{-2}$ [7]. It is clear that even

highly momentous predictions of climate models are extremely sensitive to the model's calculation of RH figures.

In combination with the sea–air properties that are numerically available from TEOS-10, the metrological definition of relative fugacity [77] offers an opportunity for a more mathematically precise formulation of the thermodynamic driving force of evaporation than the traditional Dalton equation. At marine surface conditions, this definition of RF is shown in Equation (23)

$$\psi_f(x_V, T) \equiv \frac{f_V^{\mathrm{AV}}(x_V, T)}{f_V^{\mathrm{AV,sat}}} \tag{64}$$

where the saturation fugacity, $f_V^{\mathrm{AV,sat}} \equiv f_V^{\mathrm{AV}}(x_V^{\mathrm{sat}}, T)$, is calculated either with respect to liquid water or ice Ih. For numerical implementation, Equation (64) may equivalently be expressed in terms of the chemical potentials available from the TEOS-10 SIA library [88–90]

$$R_W T \ln \psi_f(A, T, p_{\mathrm{SO}}) = \mu_W^{\mathrm{AV}}(A, T, p_{\mathrm{SO}}) - \begin{cases} g^W(T, p_{\mathrm{SO}}) \text{ for water,} \\ g^{\mathrm{Ih}}(T, p_{\mathrm{SO}}) \text{ for ice Ih,} \end{cases} \tag{65}$$

or analytically expressed, as given in the Appendices A–C, in Equations (A3), (A29) and (A37).

The marine surface RH of about 80 %rh is near saturation. Under this condition, convenient approximations of RF are available in terms of the dew-point or frost-point temperature [90]

$$R_W T \ln \psi_f \approx \left(1 - \frac{T}{T_{\mathrm{dp}}}\right) L^{\mathrm{evap}}\left(T_{\mathrm{dp}}\right) \tag{66}$$

relative to liquid water or,

$$R_W T \ln \psi_f \approx \left(1 - \frac{T}{T_{\mathrm{fp}}}\right) L^{\mathrm{subl}}\left(T_{\mathrm{fp}}\right) \tag{67}$$

relative to ice. Here, $T_{\mathrm{dp}}$ and $T_{\mathrm{fp}}$, respectively, are the dew- and frost-point temperatures of the near-surface layer of humid air, and $L^{\mathrm{evap}}$ and $L^{\mathrm{subl}}$, respectively, are the latent heats of evaporation, Equation (59), evaluated at $S = 0$, and of sublimation, Equation (60).

A special form of approximation appears slightly above the melting temperature $T_{\mathrm{mp}}$ of ice if RF needs to be evaluated with respect to liquid water, while the sample's condensation upon chilling would occur at $T_{\mathrm{fp}}$, below the freezing point as frost, that is, $T > T_{\mathrm{mp}} > T_{\mathrm{fp}}$. Then, the combination,

$$R_W T \ln \psi_f \approx \left(1 - \frac{T}{T_{\mathrm{mp}}}\right) L^{\mathrm{evap}}(T_{\mathrm{mp}}) + \left(\frac{T}{T_{\mathrm{mp}}} - \frac{T}{T_{\mathrm{fp}}}\right) L^{\mathrm{subl}}(T_{\mathrm{mp}}) \tag{68}$$

of evaporation and sublimation enthalpies should be considered. At 80% rh and below 30 °C, these simple estimates, Equations (66)–(68) agree with the exact TEOS-10 expression, Equation (65), to within 0.04 %rh [90]. Note that, here, $T_{\mathrm{dp}}$ and $T_{\mathrm{mp}}$ are properties with respect to humid air and pure water, rather than being related to seawater.

With the approximation (66) for a sufficiently warm ocean, the unbiased Dalton Equation (38), in terms of relative fugacity as a function of the dew-point temperature, may be implemented in the simple form,

$$J_W \approx -D_f \left[\frac{M_W}{M_S} S + \left(\frac{1}{T} - \frac{1}{T_{\mathrm{dp}}}\right) \frac{L^{\mathrm{evap}}\left(T_{\mathrm{dp}}\right)}{R_W}\right] \tag{69}$$

With the TEOS-10 evaporation enthalpy, Equation (59), of pure water,

$$L^{\text{evap}} = h^{\text{AV}} - A\left(\frac{\partial h^{\text{AV}}}{\partial A}\right)_{T,p} - h^{\text{W}} \approx \left(2500 - 2.5\frac{t}{°\text{C}}\right)\frac{\text{kJ}}{\text{kg}} \tag{70}$$

the approximately explicit Equation (69) for the volume flux density of liquid water by evaporation is, with the molar mass of sea salt [101], $M_{\text{S}} = 0.031\,403\,822$ kg mol$^{-1}$,

$$J_{\text{W}} \approx -D_f\left[0.02 + \left(\frac{1}{T} - \frac{1}{T_{\text{dp}}}\right)\left(1 - 0.001 \times \frac{t_{\text{dp}}}{°\text{C}}\right) \times 5417\,\text{K}\right] \tag{71}$$

Similar expressions in terms of the sublimation enthalpy apply in the vicinity of freezing temperatures.

## 6. Conclusions

The Thermodynamic Equation of Seawater 2010 (TEOS-10) is the first international geophysical standard that provides the thermodynamic properties of seawater, ice and humid air in a perfectly consistent, axiomatic way, including quantities such as entropy, enthalpy and chemical potential, which were almost unavailable before. This consistency and completeness, in combination with its unprecedented accuracy, constitutes a substantial advantage for its possible application in numerical climate models as compared to previous collections of separate property equations, which often differ from author to author and are not necessarily mutually consistent from formula to formula. Explicit equations for the TEOS-10 properties at the ocean surface are reported in Appendices A–C.

TEOS-10 provides thermodynamic equations (Equations (59) and (60)) for the enthalpies of evaporation of seawater, as well as for the sublimation of ice into humid air, including deviations from ideal gas properties. This latent heat of water vapour, when released upon condensation in clouds, provides the most important source of energy to the weather processes in the troposphere. Unfortunately, uncertainties in observing and modelling the hydrological cycle significantly exceed the observed effects of global warming in the atmosphere and the ocean, as in Equation (63).

Evaporation and sublimation rates are mostly modelled by the historical Dalton equation, which expresses the driving forces according to the difference between saturation and in situ humidity. At a constant relative humidity, as empirically observed, this form of the Dalton equation predicts increasing evaporation, caused by a globally warming atmosphere, as in Equations (4) and (5). Spuriously amplified ocean cooling, however, is inconsistent with the observed warming rate.

In terms of irreversible thermodynamics, the driving force for evaporation appears to be the difference in the chemical potentials of water in the air and the condensed phase, as in Equation (12). This difference is equivalent to the relative fugacity of humid air, shown in Equation (22), which is available from TEOS-10. This formulation avoids the putative climatological bias of the historical Dalton equation in the form of Equations (4) and (5), whose evaporation rate exponentially increases according to the saturation humidity, $q_{\text{eq}}$, which may spuriously increase along with the global temperature. Relative fugacity may be estimated from conventional relative humidity, Equation (39), which provides a ratio of humidities rather than the difference between them. Another new and computationally simple alternative for estimating relative fugacity reasonably well uses the dew-point temperature together with latent heat, in the form of a Clausius–Clapeyron formula, as in Equation (69).

The consequences of using the generalised thermodynamic driving force for the determination of the evaporation flux, especially for the aerodynamic pre-factor under diabatic conditions, will be the subject of a forthcoming study.

**Author Contributions:** R.F.: Manuscript idea and draft, elaboration of the theory; O.H.: independent verification of the theory and calculus, discussion and revision. All authors have read and agreed to the published version of the manuscript.

**Funding:** This research received no external funding.

**Institutional Review Board Statement:** Not applicable.

**Informed Consent Statement:** Not applicable.

**Data Availability Statement:** All data used are published in the cited literature, such as in open-access IAPWS and TEOS-10 documents.

**Acknowledgments:** This work contributes to the tasks of the IAPWS/SCOR/IAPSO Joint Committee on the Properties of Seawater (JCS). The contribution of O. H. was provided within the research theme 2 "Aerosols and clouds, long-term processes and trends" of the TROPOS Leibniz Institute for Tropospheric Research. The authors appreciate the helpful suggestions of the anonymous reviewers.

**Conflicts of Interest:** The authors declare no conflict of interest.

## Appendix A. Surface-Pressure Gibbs Functions of Liquid Water and Seawater

The TEOS-10 Gibbs function of seawater takes the form [5,20,25,107,108]

$$g^{\mathrm{SW}}(S,T,p) = g^{\mathrm{W}}(T,p) + g^{\mathrm{S}}(S,T,p) \tag{A1}$$

This expresses the specific Gibbs energy (per mass of seawater) as a function of absolute salinity, $S$, on the *2008 Reference-Composition Salinity Scale* [19], of absolute temperature, $T$, on the *1990 International Temperature Scale* (ITS-90, [109]), and of absolute pressure, $p$.

The pure water part, $g^{\mathrm{W}}$, of (A1) is rigorously defined by the 1995 Helmholtz function of fluid water [110], but liquid water at ambient conditions may be approximated with sufficient accuracy by the polynomial [20,107,111],

$$g^{\mathrm{W}}(T,p) = g^* \sum_{j=0}^{7} \sum_{k=0}^{6} g_{jk} \left( \frac{T - T_{\mathrm{SO}}}{T^*} \right)^j \left( \frac{p - p_{\mathrm{SO}}}{p^*} \right)^k \tag{A2}$$

Here, $g^* = 1\,\mathrm{J\,kg^{-1}}$, $T^* = 40\,\mathrm{K}$ and $p^* = 10^8\,\mathrm{Pa}$ are scaling constants. The reference point is the standard ocean with $T_{\mathrm{SO}} = 273.15\,\mathrm{K}$ and the surface pressure, $p_{\mathrm{SO}} = 101{,}325\,\mathrm{Pa}$. Of practical relevance for application at the air–sea interface, $p = p_{\mathrm{SO}}$, at Celsius temperature, $t$, are the specific Gibbs energy (rather than the full Gibbs function),

$$g^{\mathrm{W}}(T,p_{\mathrm{SO}}) = 1\,\mathrm{J\,kg^{-1}} \sum_{j=0}^{7} g_{j0} \left( \frac{t}{40\,^\circ\mathrm{C}} \right)^j \tag{A3}$$

and the pressure derivative of (A2), with the specific volume at $p = p_{\mathrm{SO}}$,

$$v^{\mathrm{W}}(T,p_{\mathrm{SO}}) = g_p^{\mathrm{W}} = 10^{-8}\,\mathrm{m^3\,kg^{-1}} \sum_{j=0}^{7} g_{j1} \left( \frac{t}{40\,^\circ\mathrm{C}} \right)^j \tag{A4}$$

In this reduced low-pressure part of TEOS-10, water appears incompressible. The coefficients $g_{jk}$ are listed in Table A1.

The temperature derivatives $g_t^{\mathrm{W}}$ and $g_{tt}^{\mathrm{W}}$ of Equation (A3) provide the specific isobaric heat capacity, $c_p^{\mathrm{W}} = -T g_{tt}^{\mathrm{W}}$, the specific entropy, $\eta^{\mathrm{W}} = -g_t^{\mathrm{W}}$, and the specific enthalpy, $h^{\mathrm{W}} = g^{\mathrm{W}} - T g_t^{\mathrm{W}}$, of pure liquid water,

$$h^{\mathrm{W}} = 1\,\mathrm{J\,kg^{-1}} \left[ g_{00} - \frac{273.15}{40} g_{10} + \sum_{j=2}^{7} g_{j0} \left( \frac{t}{40\,^\circ\mathrm{C}} - j\frac{T}{40\,\mathrm{K}} \right) \left( \frac{t}{40\,^\circ\mathrm{C}} \right)^{j-1} \right] \tag{A5}$$

The chemical potential of liquid water is given by $\mu_{\mathrm{W}}^{\mathrm{W}} = g^{\mathrm{W}}$.

**Table A1.** Coefficients of Equations (A3) and (A4).

| $j$ | $g_{j0}$ of Equation (A3) | $g_{j1}$ of Equation (A4) |
|---|---|---|
| 0 | $0.101\ 342\ 743\ 139\ 672 \times 10^3$ | $0.100\ 015\ 695\ 367\ 145 \times 10^6$ |
| 1 | $0.590\ 578\ 348\ 518\ 236 \times 10$ | $-0.270\ 983\ 805\ 184\ 062 \times 10^3$ |
| 2 | $-0.123\ 577\ 859\ 330\ 390 \times 10^5$ | $0.145\ 503\ 645\ 404\ 680 \times 10^4$ |
| 3 | $0.736\ 741\ 204\ 151\ 612 \times 10^3$ | $-0.672\ 507\ 783\ 145\ 070 \times 10^3$ |
| 4 | $-0.148\ 185\ 936\ 433\ 658 \times 10^3$ | $0.397\ 968\ 445\ 406\ 972 \times 10^3$ |
| 5 | $0.580\ 259\ 125\ 842\ 571 \times 10^2$ | $-0.194\ 618\ 310\ 617\ 595 \times 10^3$ |
| 6 | $-0.189\ 843\ 846\ 514\ 172 \times 10^2$ | $0.635\ 113\ 936\ 641\ 785 \times 10^2$ |
| 7 | $0.305\ 081\ 646\ 487\ 967 \times 10$ | $-0.963\ 108\ 119\ 393\ 062 \times 10$ |

Over the last century, the salt content of seawater was measured using different methods and expressed on different scales. There are two ways of dealing with deviating numerical values that represent the same physical matter; they can either be considered as the same salinity, expressed with respect to different units, or they can be considered as different quantities. In TEOS-10, several salinities are distinguished [104,112–114]; notably, »Practical Salinity« is distinguished from »Absolute Salinity«. Practical Salinity, $S_P$, is computed from measured electrical conductivity, as defined by the 1978 *Practical Salinity Scale*, PSS-78 [115,116]. Absolute Salinity, $S_A$, is computed from measured density, as defined by the 2008 *Reference-Composition Salinity Scale* [19]. Common oceanographic instruments return $S_P$ values, which are stored in marine databases. In contrast, the equations of TEOS-10 are expressed in terms of $S_A$, which also considers non-dissociated solutes such as silicate.

In the case of *IAPSO Standard Seawater* with the reference composition, there is a fixed relation between $S_P$ and $S_A$. For this reason, a single salinity variable $S$ will be used here, which may use either practical salinity unit (psu) or mass fraction units (g kg$^{-1}$, kg kg$^{-1}$ or unitless). The conversion formula is [19]

$$1\ \text{psu} = 1.004\ 715\ \text{g kg}^{-1} = 0.001\ 004\ 715\ \text{kg kg}^{-1} = 0.001\ 004\ 715 \tag{A6}$$

Note that the developers of PSS-78 strongly discouraged the use of »psu« [102]; however, regardless, many oceanography authors use psu for convenience and clarity, such as [72], to avoid confusion with other, earlier unitless salinity values.

The TEOS-10 equation of the saline part of the Gibbs function (A1) is [20,25,107,108]:

$$g^S(S,T,p) = g^* \sum_{j=0}^{6} \sum_{k=0}^{5} \left[ \frac{1}{2} g_{1jk} \left( \frac{S}{S^*} \right) \ln \left( \frac{S}{S^*} \right) \right. \\ \left. + \sum_{i=2}^{7} g_{ijk} \left( \frac{S}{S^*} \right)^{i/2} \right] \left( \frac{T-T_{SO}}{T^*} \right)^j \left( \frac{p-p_{SO}}{p^*} \right)^k \tag{A7}$$

Salinity is scaled here as $S^* = 40\ \text{psu} = 40.188\ 617\ \text{g kg}^{-1} = 0.040\ 188\ 617$. The leading terms within the squared brackets have clear meanings in electrolyte theory and provide the dominant contributions. The logarithmic term results from the theory of ideal solutions, $i = 2$ describes the ion-water interaction energy, $i = 3$ is Debye's limiting law, and $i = 4$ follows from short-range ion–ion interaction forces. Mathematically, the logarithmic term gives rise to Raoult's law of vapour-pressure lowering and freezing-point depression.

At the ocean surface, $p = p_{SO}$, Equation (A7) is reduced to

$$g^S(S,T,p_{SO}) = 1\ \text{J kg}^{-1} \sum_{j=0}^{6} \left[ \frac{1}{2} g_{1j0} \left( \frac{S}{S^*} \right) \ln \left( \frac{S}{S^*} \right) + \sum_{i=2}^{7} g_{ij0} \left( \frac{S}{S^*} \right)^{i/2} \right] \left( \frac{t}{40\ °\text{C}} \right)^j \tag{A8}$$

The pressure derivative of (A7) at the surface is

$$g_p^{\mathrm{S}}(S,T,p_{\mathrm{SO}}) = 10^{-8}\ \mathrm{m^3\,kg^{-1}} \sum_{j=0}^{4}\sum_{i=2}^{5} g_{ij1}\left(\frac{S}{S^*}\right)^{i/2}\left(\frac{t}{40\ ^\circ\mathrm{C}}\right)^{j} \qquad (A9)$$

For theoretical reasons, the logarithmic term of Equation (A7) does not depend on pressure. The coefficients of Equations (A8) and (A9) are listed in Table A2.

The temperature derivatives $g_t^{\mathrm{SW}}$ and $g_{tt}^{\mathrm{SW}}$ of Equation (A1) provide the specific isobaric heat capacity, $c_P^{\mathrm{SW}} = -T g_{tt}^{\mathrm{SW}}$, the specific entropy, $\eta^{\mathrm{SW}} = -g_t^{\mathrm{SW}}$, and the specific enthalpy, $h^{\mathrm{SW}} = g^{\mathrm{SW}} - T g_t^{\mathrm{SW}}$, of seawater. The chemical potential of water in seawater is given by

$$\mu_{\mathrm{W}}^{\mathrm{SW}} = g^{\mathrm{SW}} - S\left(\frac{\partial g^{\mathrm{SW}}}{\partial S}\right)_{T,p} = g^{\mathrm{W}} + g^{\mathrm{S}} - S\left(\frac{\partial g^{\mathrm{S}}}{\partial S}\right)_{T,p} \qquad (A10)$$

and the specific enthalpy is

$$h^{\mathrm{SW}} = g^{\mathrm{SW}} - T\left(\frac{\partial g^{\mathrm{SW}}}{\partial T}\right)_{S,p} \qquad (A11)$$

**Table A2.** Coefficients of Equations (A8) and (A9).

| $i$ | $j$ | $g_{ij0}$ of Equation (A8) | $g_{ij1}$ of Equation (A9) |
|---|---|---|---|
| 1 | 0 | 0.581 281 456 626 732 × $10^4$ | - |
| 2 | 0 | 0.141 627 648 484 197 × $10^4$ | −0.331 049 154 044 839 × $10^4$ |
| 3 | 0 | −0.243 214 662 381 794 × $10^4$ | 0.199 459 603 073 901 × $10^3$ |
| 4 | 0 | 0.202 580 115 603 697 × $10^4$ | −0.547 919 133 532 887 × $10^2$ |
| 5 | 0 | −0.109 166 841 042 967 × $10^4$ | 0.360 284 195 611 086 × $10^2$ |
| 6 | 0 | 0.374 601 237 877 840 × $10^3$ | - |
| 7 | 0 | −0.485 891 069 025 409 × $10^2$ | - |
| 1 | 1 | 0.851 226 734 946 706 × $10^3$ | - |
| 2 | 1 | 0.168 072 408 311 545 × $10^3$ | 0.729 116 529 735 046 × $10^3$ |
| 3 | 1 | −0.493 407 510 141 682 × $10^3$ | −0.175 292 041 186 547 × $10^3$ |
| 4 | 1 | 0.543 835 333 000 098 × $10^3$ | −0.226 683 558 512 829 × $10^2$ |
| 5 | 1 | −0.196 028 306 689 776 × $10^3$ | - |
| 6 | 1 | 0.367 571 622 995 805 × $10^2$ | - |
| 2 | 2 | 0.880 031 352 997 204 × $10^3$ | −0.860 764 303 783 977 × $10^3$ |
| 3 | 2 | −0.430 664 675 978 042 × $10^2$ | 0.383 058 066 002 476 × $10^3$ |
| 4 | 2 | −0.685 572 509 204 491 × $10^2$ | - |
| 2 | 3 | −0.225 267 649 263 401 × $10^3$ | 0.694 244 814 133 268 × $10^3$ |
| 3 | 3 | −0.100 227 370 861 875 × $10^2$ | −0.460 319 931 801 257 × $10^3$ |
| 4 | 3 | 0.493 667 694 856 254 × $10^2$ | - |
| 2 | 4 | 0.914 260 447 751 259 × $10^2$ | −0.297 728 741 987 187 × $10^3$ |
| 3 | 4 | 0.875 600 661 808 945 | 0.234 565 187 611 355 × $10^3$ |
| 4 | 4 | −0.171 397 577 419 788 × $10^2$ | - |
| 2 | 5 | −0.216 603 240 875 311 × $10^2$ | - |
| 4 | 5 | 0.249 697 009 569 508 × 10 | - |
| 2 | 6 | 0.213 016 970 847 183 × 10 | - |

**Appendix B. Virial Gibbs Function of Humid Air**

In TEOS-10, the equation of state of humid air is defined as a Helmholtz function [24,117],

$$f^{\text{AV}}(A, T, \rho) = A f^{\text{A}}(T, A\rho) + (1 - A) f^{\text{V}}(T, (1 - A)\rho) + f^{\text{mix}}(A, T, \rho) \tag{A12}$$

Here, $A$ is the mass fraction of dry air, $T$ is the absolute ITS-90 temperature, $\rho$ is the mass density, $f^{\text{A}}$ is the Helmholtz function of dry air [118], $f^{\text{V}}$ is the IAPWS-95 Helmholtz function of water vapour [110,119], and $f^{\text{mix}}$ is the virial formula for air–water interactions. For atmospheric low-pressure applications, the functions $f^{\text{A}}$, $f^{\text{V}}$ and $f^{\text{mix}}$ may be approximated by their lowest-order virial expansions [76,120]:

$$f^{\text{A}}\left(T, \rho^{\text{A}}\right) \approx f_0^{\text{A}}(T) + \frac{R^{\text{L}} T}{M_{\text{A}}} \ln \frac{\rho^{\text{A}}}{\rho^*} + \frac{R^{10} T}{M_{\text{A}}^2} \rho^{\text{A}} B^{\text{AA}}(T) \tag{A13}$$

$$f^{\text{V}}\left(T, \rho^{\text{V}}\right) \approx f_0^{\text{V}}(T) + \frac{R^{95} T}{M_{\text{W}}} \ln \frac{\rho^{\text{V}}}{\rho^*} + \frac{R^{10} T}{M_{\text{W}}^2} \rho^{\text{V}} B^{\text{WW}}(T) \tag{A14}$$

$$f^{\text{mix}}(A, T, \rho) \approx 2A(1 - A) \frac{R^{10} T}{M_{\text{A}} M_{\text{W}}} \rho B^{\text{AW}}(T) \tag{A15}$$

Here, $\rho^{\text{A}} \equiv A\rho$ and $\rho^{\text{V}} \equiv (1 - A)\rho$ are the partial mass densities, respectively, of dry air and water vapour. The formal unit density is $\rho^* = 1 \text{ kg m}^{-3}$. The molar gas constant used in 2010 was $R^{10} = 8.314\,472 \text{ J mol}^{-1} \text{ K}^{-1}$. The molar masses of dry air and pure water, respectively, are $M_{\text{A}} = 0.028\,965\,46 \text{ kg mol}^{-1}$ and $M_{\text{W}} = 0.018\,015\,268 \text{ kg mol}^{-1}$. The new functions that were used are defined below.

The ideal gas thermal part $f_0^{\text{A}}$ of the Helmholtz function for dry air [118], adjusted to the TEOS-10 reference state of humid air, is [120]:

$$f_0^{\text{A}}(T) = \frac{R^{\text{L}} T}{M_{\text{A}}} \left\{ \ln \frac{\rho^*}{\rho_{\text{A}}^*} + \sum_{i=1}^{5} n_i^{\text{A}} \tau^{i-4} + n_6^{\text{A}} \tau^{1.5} \right.$$
$$\left. + n_7^{\text{A}} \ln \tau + \sum_{i=8}^{9} n_i^{\text{A}} \ln\left(1 - e^{-n_{i+3}^{\text{A}} \tau}\right) + n_{10}^{\text{A}} \ln\left(\frac{2}{3} + e^{n_{13}^{\text{A}} \tau}\right) \right\} \tag{A16}$$

Used by Lemmon et al. [118], the molar gas constant is $R^{\text{L}} = 8.314\,51 \text{ J mol}^{-1} \text{ K}^{-1}$. The reducing density is $\rho_{\text{A}}^* = 10,447.7 \text{ mol m}^{-3} \times M_{\text{A}}$; the temperature variable is $\tau = (132.6312 \text{ K})/T$.

The ideal gas thermal part $f_0^{\text{V}}$ of the Helmholtz function for water vapour is [119]

$$f_0^{\text{V}}(T) = \frac{R^{95} T}{M_{\text{W}}} \left\{ -\ln 322 + n_1^{\text{V}} + n_2^{\text{V}} \tau + n_3^{\text{V}} \ln \tau + \sum_{i=4}^{8} n_i^{\text{V}} \ln\left(1 - e^{-n_{i+5}^{\text{V}} \tau}\right) \right\} \tag{A17}$$

The molar gas constant used by Wagner and Pruß [110] is $R^{95} = R_{\text{W}}^{95} \times M_{\text{W}}$, $R_{\text{W}}^{95} = 461.518\,05 \text{ J kg}^{-1} \text{ K}^{-1}$. The temperature variable is $\tau = (647.096 \text{ K})/T$. The temperature dependence of Equations (A16) and (A17) describes the two ideal gas heat capacities and enthalpies over a wide range of temperatures. The coefficients are listed in Table A3.

**Table A3.** Coefficients of Equations (A16) and (A17). Coefficients $n_4^A$ and $n_3^A$ differ from those of Lemmon et al. [118], as they are adjusted to the geophysical reference state at 0 °C and 101,325 Pa.

| $i$ | $n_i^A$ of Equation (A16) | $n_i^V$ of Equation (A17) |
|---|---|---|
| 1 | $0.605\ 7194 \times 10^{-7}$ | $-0.832\ 044\ 648\ 374\ 969 \times 10$ |
| 2 | $-0.210\ 274\ 769 \times 10^{-4}$ | $0.668\ 321\ 052\ 759\ 323 \times 10$ |
| 3 | $-0.158\ 860\ 716 \times 10^{-3}$ | $0.300\ 632 \times 10$ |
| 4 | $0.974\ 502\ 517\ 439\ 48 \times 10$ | $0.124\ 36 \times 10^{-1}$ |
| 5 | $0.100\ 986\ 147\ 428\ 912 \times 10^2$ | $0.973\ 15$ |
| 6 | $-0.195\ 363\ 42 \times 10^{-3}$ | $0.127\ 95 \times 10$ |
| 7 | $0.249\ 088\ 8032 \times 10$ | $0.969\ 56$ |
| 8 | $0.791\ 309\ 509$ | $0.248\ 73$ |
| 9 | $0.212\ 236\ 768$ | $0.128\ 728\ 967 \times 10$ |
| 10 | $-0.197\ 938\ 904$ | $0.353\ 734\ 222 \times 10$ |
| 11 | $0.253\ 6365 \times 10^2$ | $0.774\ 073\ 708 \times 10$ |
| 12 | $0.169\ 0741 \times 10^2$ | $0.924\ 437\ 796 \times 10$ |
| 13 | $0.873\ 1279 \times 10^2$ | $0.275\ 075\ 105 \times 10^2$ |

In Equations (A13)–(A15), the second virial coefficients of air–air, water–water and air–water molecular interactions, respectively, are [76]

$$B^{AA}(T) = \frac{1}{\rho_{AA}^*} \sum_{i=1}^{6} a_i \tau^{b_i} \tag{A18}$$

with $\rho_{AA}^* = 10,447.7 \text{ mol m}^{-3}$, $\tau = (132.6312 \text{ K})/T$,

$$B^{WW}(T) = \frac{1}{\rho_{WW}^*} \left\{ \sum_{i=1}^{7} a_i \tau^{b_i} + \sum_{i=8}^{9} a_i \left[ 0.2 + (1.32 - \tau)^2 \right]^{b_i} e^{-c_i - d_i(\tau-1)^2} \right\} \tag{A19}$$

with $\rho_{WW}^* = \left(322 \text{ kg m}^{-3}\right)/M_W$, $\tau = (647.096 \text{ K})/T$, and [121]

$$B^{AW}(T) = \frac{1}{\rho_{AW}^*} \sum_{i=1}^{3} a_i \tau^{b_i} \tag{A20}$$

with $\rho_{AW}^* = 10^6 \text{ mol m}^{-3}$, $\tau = T/(100 \text{ K})$. The remaining coefficients are listed in Tables A4–A6.

**Table A4.** Coefficients of $B^{AA}(T)$, Equation (A18).

| $i$ | $a_i$ | $b_i$ |
|---|---|---|
| 1 | $0.118\ 160\ 747\ 229$ | $0$ |
| 2 | $0.713\ 116\ 392\ 079$ | $0.33$ |
| 3 | $-0.161\ 824\ 192\ 067 \times 10$ | $1.01$ |
| 4 | $-0.101\ 365\ 037\ 912$ | $1.6$ |
| 5 | $-0.146\ 629\ 609\ 713$ | $3.6$ |
| 6 | $0.148\ 287\ 891\ 978 \times 10^{-1}$ | $3.5$ |

**Table A5.** Coefficients of $B^{\mathrm{WW}}(T)$, Equation (A19).

| $i$ | $a_i$ | $b_i$ | $c_i$ | $d_i$ |
|---|---|---|---|---|
| 1 | $0.125\,335\,479\,355\,23 \times 10^{-1}$ | $-0.5$ | - | - |
| 2 | $0.789\,576\,347\,228\,28 \times 10$ | $0.875$ | - | - |
| 3 | $-0.878\,032\,033\,035\,61 \times 10$ | $1$ | - | - |
| 4 | $-0.668\,565\,723\,079\,65$ | $4$ | - | - |
| 5 | $0.204\,338\,109\,509\,65$ | $6$ | - | - |
| 6 | $-0.662\,126\,050\,396\,87 \times 10^{-4}$ | $12$ | - | - |
| 7 | $-0.107\,936\,009\,089\,32$ | $7$ | - | - |
| 8 | $-0.148\,746\,408\,567\,24$ | $0.85$ | $28$ | $700$ |
| 9 | $0.318\,061\,108\,784\,44$ | $0.95$ | $32$ | $800$ |

**Table A6.** Coefficients of $B^{\mathrm{AW}}(T)$, Equation (A20).

| $i$ | $a_i$ | $b_i$ |
|---|---|---|
| 1 | $0.665\,687 \times 10^2$ | $-0.237$ |
| 2 | $-0.238\,834 \times 10^3$ | $-1.048$ |
| 3 | $-0.176\,755 \times 10^3$ | $-3.183$ |

With the abbreviation,

$$f_0^{\mathrm{AV}}(A,T) \equiv A\left[f_0^{\mathrm{A}}(T) + \frac{R^{\mathrm{L}}T}{M_{\mathrm{A}}}\ln A\right] + (1-A)\left[f_0^{\mathrm{V}}(T) + \frac{R^{95}T}{M_{\mathrm{W}}}\ln(1-A)\right] \quad (A21)$$

using the mean molar mass,

$$M_{\mathrm{AW}}(A) \equiv R\left\{\frac{R^{\mathrm{L}}A}{M_{\mathrm{A}}} + \frac{R^{95}(1-A)}{M_{\mathrm{W}}}\right\}^{-1} \quad (A22)$$

and the mixture virial coefficient,

$$B(A,T) \equiv \frac{R^{10}}{R}M_{\mathrm{AW}}^2\left\{\frac{A^2}{M_{\mathrm{A}}^2}B^{\mathrm{AA}} + \frac{(1-A)^2}{M_{\mathrm{W}}^2}B^{\mathrm{WW}} + 2\frac{A(1-A)}{M_{\mathrm{A}}M_{\mathrm{W}}}B^{\mathrm{AW}}\right\} \quad (A23)$$

from the Helmholtz function, Equation (A12), takes the form,

$$f^{\mathrm{AV}}(A,T,\rho) = f_0^{\mathrm{AV}}(A,T) + \frac{RT}{M_{\mathrm{AW}}}\ln\frac{\rho}{\rho^*} + \frac{RT}{M_{\mathrm{AW}}^2}\rho B \quad (A24)$$

and the pressure, $p(A,T,\rho) = \rho^2\left(\partial f^{\mathrm{AV}}/\partial\rho\right)_{A,T}$, is derived,

$$\frac{p}{RT} = \frac{\rho}{M_{\mathrm{AW}}} + \left(\frac{\rho}{M_{\mathrm{AW}}}\right)^2 B \quad (A25)$$

The final exact value of the molar gas constant, $R = 8.314\,462\,618\,153\,24$ J mol$^{-1}$ K$^{-1}$, was introduced for convenience [98].

The density can be analytically caluclated using quadratic Equation (A25),

$$\rho(A,T,p) = \frac{M_{\mathrm{AW}}(A)}{2B(A,T)}\left(\sqrt{1 + \frac{4pB(A,T)}{RT}} - 1\right) \quad (A26)$$

Making use of Equation (A26), the virial Helmholtz function, Equation (A24), may be converted into the related, practically more convenient, Gibbs function, $g^{AV}(A, T, p) = f^{AV} + p\rho^{-1}$,

$$g^{AV} = f_0^{AV}(A, T) + \frac{RT}{M_{AW}} \left[ \ln \frac{M_{AW}}{2B\rho^*} + \ln \left( \sqrt{1 + \frac{4pB}{RT}} - 1 \right) + \sqrt{1 + \frac{4pB}{RT}} \right] \quad \text{(A27)}$$

This is the mathematically exact Gibbs function, $g^{AV}(A, T, p)$, for humid air, in terms of the second virial coefficients, Equation (A24), as an approximation of the related full Helmholtz function provided by TEOS-10. Up to linear terms, the series expansion with respect to $B$, Equation (A27), is reduced to

$$g^{AV} = f_0^{AV}(A, T) + \frac{RT}{M_{AW}} \left( 1 + \ln \frac{M_{AW} p}{\rho^* RT} \right) + \frac{pB}{M_{AW}} + O\left(B^2\right) \quad \text{(A28)}$$

Derived from this linearised Gibbs function, Equation (A28) provides the chemical potential of water in humid air

$$\mu_W^{AV} = g^{AV} - A \left( \frac{\partial g^{AV}}{\partial A} \right)_{T,p} = \mu_W^{id}(T, x_V p) + \frac{R^{95} T}{M_W} \ln \frac{f_V^{AV}(A, T, p)}{x_V p} \quad \text{(A29)}$$

which consists of an ideal-gas contribution,

$$\mu_W^{id}(T, x_V p) = f_0^V(T) + \frac{R^{95} T}{M_W} \left[ 1 + \ln \frac{x_V p M_W}{\rho^* R^{95} T} - \ln(1 - \varepsilon_V) \right] \quad \text{(A30)}$$

and a virial correction for molecular interaction, expressed by the fugacity, $f_V^{AV}$,

$$\frac{R^{95} T}{M_W} \ln \frac{f_V^{AV}(A, T, p)}{x_V p} = \frac{R^{10} R^L}{R^2} \frac{p}{M_W} \left[ x_V(2 - x_V) B^{WW} + (1 - x_V)^2 \left( 2B^{AW} - B^{AA} \right) + \varepsilon_f \right] \quad \text{(A31)}$$

Here, the mole fraction of water vapour is given by Equation (15),

$$x_V(A) = \left[ 1 + \frac{A M_W}{(1 - A) M_A} \right]^{-1} \quad \text{(A32)}$$

and the numerically negligible errors, $\varepsilon_V, \varepsilon_f$, caused by the varying obsolete values of the molar gas constant, are included in Equation (A30),

$$\varepsilon_V = \left( 1 - \frac{R^L}{R^{95}} \right) (1 - x_V) \quad \text{(A33)}$$

and Equation (A31),

$$\varepsilon_f = \left( \frac{R^{95}}{R^L} - 1 \right) \left[ (x_V)^2 B^{WW} - (1 - x_V)^2 B^{AA} \right] \quad \text{(A34)}$$

Assuming equal values of the previous gas constants, $R^{95} = R^L = R^{10} = R$, from Equation (A31), the regular form of the ideal gas chemical potential of water can be derived,

$$\mu_W^{id}(T, x_V p) = f_0^V(T) + \frac{RT}{M_W} \left( 1 + \ln \frac{x_V p M_W}{\rho^* RT} \right) \quad \text{(A35)}$$

as well as its fugacity in a binary air–water gas mixture [73,76],

$$f_V^{AV}(A, T, p) = x_V p \exp \left\{ \left[ x_V(2 - x_V) B^{WW} + (1 - x_V)^2 \left( 2B^{AW} - B^{AA} \right) \right] \frac{p}{RT} \right\} \quad \text{(A36)}$$

**Appendix C. Surface-Pressure Gibbs Function of Ice Ih**

The TEOS-10 Gibbs function of ambient hexagonal ice, termed »ice Ih«, as compared to cubic ice Ic or ices II, III, etc. [122], is expressed as the real part (Re) of a complex function value. This complex notation has no physical reasons but permits a mathematically compact formula. At the sea-surface pressure, $p = p_{SO}$, the specific Gibbs energy of ice Ih is [23,123],

$$g^{Ih}(T, p_{SO}) = g_{00} - \eta_0 T$$
$$+ T_t \, \text{Re} \left\{ \sum_{k=1}^{2} r_{k0} \left[ \left( t_k - \frac{T}{T_t} \right) \ln \left( t_k - \frac{T}{T_t} \right) + \left( t_k + \frac{T}{T_t} \right) \ln \left( t_k + \frac{T}{T_t} \right) - 2 t_k \ln t_k - \frac{(T/T_t)^2}{t_k} \right] \right\} \quad \text{(A37)}$$

and the specific volume is,

$$v^{Ih}(T, p_{SO}) = \frac{g_{01}}{p_t} + T_t \, \text{Re} \left\{ \frac{r_{21}}{p_t} \left[ \left( t_2 - \frac{T}{T_t} \right) \ln \left( t_2 - \frac{T}{T_t} \right) + \left( t_2 + \frac{T}{T_t} \right) \ln \left( t_2 + \frac{T}{T_t} \right) - 2 t_2 \ln t_2 - \frac{(T/T_t)^2}{t_2} \right] \right\} \quad \text{(A38)}$$

Here, the triple-point temperature is $T_t = 273.16$ K, and $p_t = 611.657$ Pa is the triple-point pressure. The remaining coefficients are reported in Table A7. In the simplified form, ice Ih is approximated as incompressible. The arbitrary constants $g_{00}$, $s_0$ are defined by TEOS-10 reference states consistent with the Gibbs functions of liquid water, seawater and humid air. However, these do not obey the third law of vanishing entropy at the zero point. This choice has no effect on any measurable properties.

**Table A7.** Coefficients of the TEOS-10 Gibbs function of ice Ih in surface pressure approximation; Equations (A37) and (A38).

| Coefficient | Real Part | Imaginary Part | Unit |
|:---:|:---:|:---:|:---:|
| $g_{00}$ | $-0.632\,020\,233\,449\,497 \times 10^6$ | - | J kg$^{-1}$ |
| $g_{01}$ | $0.655\,022\,213\,658\,955$ | - | J kg$^{-1}$ |
| $\eta_0$ | $-0.332\,733\,756\,492\,168 \times 10^4$ | - | J kg$^{-1}$K$^{-1}$ |
| $t_1$ | $0.368\,017\,112\,855\,051 \times 10^{-1}$ | $0.510\,878\,114\,959\,572 \times 10^{-1}$ | |
| $t_2$ | $0.337\,315\,741\,065\,416$ | $0.335\,449\,415\,919\,309$ | |
| $r_{10}$ | $0.447\,050\,716\,285\,388 \times 10^2$ | $0.656\,876\,847\,463\,481 \times 10^2$ | J kg$^{-1}$K$^{-1}$ |
| $r_{20}$ | $-0.725\,974\,574\,329\,220 \times 10^2$ | $-0.781\,008\,427\,112\,870 \times 10^2$ | J kg$^{-1}$K$^{-1}$ |
| $r_{21}$ | $-0.557\,107\,698\,030\,123 \times 10^{-4}$ | $0.464\,578\,634\,580\,806 \times 10^{-4}$ | J kg$^{-1}$K$^{-1}$ |

The chemical potential of ice is

$$\mu_W^{Ih} = g^{Ih} \quad \text{(A39)}$$

And the specific enthalpy is

$$h^{Ih} = g^{Ih} - T \left( \frac{\partial g^{Ih}}{\partial T} \right)_p = g_{00} + \text{Re} \left\{ \sum_{k=1}^{2} r_{k0} t_k T_t \left[ \ln \left( 1 - \left( \frac{T}{t_k T_t} \right)^2 \right) + \left( \frac{T}{t_k T_t} \right)^2 \right] \right\} \quad \text{(A40)}$$

## Appendix D. Nomenclature

| Symbol | Quantity | SI Unit | Equation |
|---|---|---|---|
| $A$ | Mass fraction of dry air in humid air, $A = 1 - q$ | kg kg$^{-1}$ | (10) |
| $a_i$ | Coefficients of virial coefficients | 1 | (A18) |
| $B$ | Mixture virial coefficient | m$^3$ mol$^{-1}$ | (A23) |
| $B^{AA}$ | 2nd virial coefficient of air–air interaction | m$^3$ mol$^{-1}$ | (A13) |
| $B^{AW}$ | 2nd virial coefficient of air–water interaction | m$^3$ mol$^{-1}$ | (A15) |
| $B^{WW}$ | 2nd virial coefficient of water–water interaction | m$^3$ mol$^{-1}$ | (A14) |
| $b_i$ | Coefficients of virial coefficients | 1 | (A18) |
| $C_L$ | Latent heat transfer coefficient [18], $C_L = 1.2 \times 10^{-3}$ | 1 | (35) |
| $C_p^{SA}$ | Isobaric heat capacity of sea air | J K$^{-1}$ | (57) |
| $c_p^{Ih}$ | Specific isobaric heat capacity of ice Ih | J kg$^{-1}$ K$^{-1}$ | (57) |
| $c_p^{AV}$ | Specific isobaric heat capacity of humid air | J kg$^{-1}$ K$^{-1}$ | (57) |
| $c_p^{SW}$ | Specific isobaric heat capacity of seawater | J kg$^{-1}$ K$^{-1}$ | (57) |
| $c_p^{W}$ | Specific isobaric heat capacity of liquid water | J kg$^{-1}$ K$^{-1}$ | (A5) |
| $c_i$ | Coefficients of virial coefficients | 1 | (A19) |
| $D$ | Dalton coefficient | m s$^{-1}$ | (1) |
| $D_e$ | Vapour-pressure-based transfer coefficient | s m$^{-1}$ | (29) |
| $D_f$ | Fugacity-based transfer coefficient | kg m$^{-2}$ s$^{-1}$ | (19) |
| $D_q$ | Humidity-based transfer coefficient | kg m$^{-2}$ s$^{-1}$ | (34) |
| $d_i$ | Coefficients of virial coefficients | 1 | (A19) |
| $e$ | Vapour pressure | Pa | (27) |
| $e^{sat}$ | Saturation vapour pressure | Pa | (27) |
| $F^{SA}$ | Helmholtz energy of sea air | J | (40) |
| $F^{SA,id}$ | Ideal gas Helmholtz energy of sea air | J | (40) |
| $f^{A}$ | Specific Helmholtz energy of dry air | J kg$^{-1}$ | (A12) |
| $f_0^{A}$ | Thermal ideal-gas Helmholtz energy of dry air | J kg$^{-1}$ | (A13) |
| $f^{AV}$ | Specific Helmholtz energy of humid air | J kg$^{-1}$ | (42) |
| $f_0^{AV}$ | Thermal ideal-gas Helmholtz energy of humid air | J kg$^{-1}$ | (A21) |
| $f^{Ih}$ | Specific Helmholtz energy of ice Ih | J kg$^{-1}$ | (42) |
| $f^{mix}$ | Specific Helmholtz energy of air–water interaction | J kg$^{-1}$ | (A12) |
| $f^{SW}$ | Specific Helmholtz energy of seawater | J kg$^{-1}$ | (42) |
| $f^{V}$ | Specific Helmholtz energy of water vapour | J kg$^{-1}$ | (A12) |
| $f_0^{V}$ | Thermal ideal gas Helmholtz energy of water vapour | J kg$^{-1}$ | (A14) |
| $f_V^{AV}$ | Fugacity of water vapour in humid air | Pa | (13) |
| $f_V^{AV,sat}$ | Saturation fugacity of water vapour in humid air | Pa | (64) |
| $f_W^{SW}$ | Fugacity of water in seawater | Pa | (14) |
| $G^{AV}$ | Gibbs energy of humid air | J | (43) |

| Symbol | Quantity | SI Unit | Equation |
|---|---|---|---|
| $G^{\text{Ih}}$ | Gibbs energy of ice Ih | J | (43) |
| $G^{\text{SA}}$ | Gibbs energy of sea air | J | (43) |
| $G^{\text{SW}}$ | Gibbs energy of seawater | J | (43) |
| $g^{\text{AV}}$ | Specific Gibbs energy of humid air | $\text{J kg}^{-1}$ | (45) |
| $g^{\text{Ih}}$ | Specific Gibbs energy of ice Ih | $\text{J kg}^{-1}$ | (45) |
| $g_{jk}$ | Coefficients of the Gibbs function of liquid water | 1 | (A2) |
| $g_{jk}$ | Coefficients of the Gibbs function of ice Ih | $\text{J kg}^{-1}$ | (A37) |
| $g_{ijk}$ | Coefficients of the saline Gibbs function | 1 | (A7) |
| $g^{\text{S}}$ | Saline part of the specific Gibbs energy of seawater | $\text{J kg}^{-1}$ | (A1) |
| $g^{\text{SW}}$ | Specific Gibbs energy of seawater | $\text{J kg}^{-1}$ | (45) |
| $g^{\text{W}}$ | Specific Gibbs energy of liquid water | $\text{J kg}^{-1}$ | (A1) |
| $g^{*}$ | Scaling specific Gibbs energy, $g^{*} = 1\,\text{J kg}^{-1}$ | $\text{J kg}^{-1}$ | (A2) |
| $H^{\text{AV}}$ | Enthalpy of humid air | J | (53) |
| $H^{\text{Ih}}$ | Enthalpy of ice Ih | J | (53) |
| $H^{\text{SA}}$ | Enthalpy of sea air | J | (53) |
| $H^{\text{SW}}$ | Enthalpy of seawater | J | (53) |
| $h^{\text{A}}$ | Specific enthalpy of dry air | $\text{J kg}^{-1}$ | (61) |
| $h^{\text{AV}}$ | Specific enthalpy of humid air | $\text{J kg}^{-1}$ | (55) |
| $h^{\text{AV, id}}$ | Specific enthalpy of ideal gas humid air | $\text{J kg}^{-1}$ | (61) |
| $h^{\text{Ih}}$ | Specific enthalpy of ice Ih | $\text{J kg}^{-1}$ | (56) |
| $h^{\text{SW}}$ | Specific enthalpy of seawater | $\text{J kg}^{-1}$ | (54) |
| $h^{\text{V}}$ | Specific enthalpy of water vapour | $\text{J kg}^{-1}$ | (61) |
| $h^{\text{W}}$ | Specific enthalpy of liquid water | $\text{J kg}^{-1}$ | (70) |
| $J_k$ | Irreversible Onsager flux | various | (6) |
| $J_{\text{W}}$ | Evaporation mass-flux density | $\text{kg m}^{-2}\,\text{s}^{-1}$ | (1) |
| $k_{\text{B}}$ | Boltzmann constant | $\text{J K}^{-1}$ | (40) |
| $L$ | Specific evaporation enthalpy | $\text{J kg}^{-1}$ | (1) |
| $L^{\text{evap}}$ | Specific evaporation enthalpy of liquid water | $\text{J kg}^{-1}$ | (57) |
| $L^{\text{subl}}$ | Specific sublimation enthalpy of ice Ih | $\text{J kg}^{-1}$ | (57) |
| $M_{\text{A}}$ | Molar mass of dry air, $M_{\text{A}} = 0.028\,965\,46\,\text{kg mol}^{-1}$ | $\text{kg mol}^{-1}$ | (15) |
| $M_{\text{AW}}$ | Molar mass of humid air | $\text{kg mol}^{-1}$ | (A22) |
| $M_{\text{S}}$ | Molar mass of dissolved sea salt, $M_{\text{S}} = 0.031\,403\,822\,\text{kg mol}^{-1}$ | $\text{kg mol}^{-1}$ | (16) |
| $M_{\text{W}}$ | Molar mass of water, $M_{\text{W}} = 0.018\,015\,268\,\text{kg mol}^{-1}$ | $\text{kg mol}^{-1}$ | (13) |
| $m^{\text{A}}$ | Mass of dry air | kg | (40) |
| $m^{\text{Ih}}$ | Mass of ice Ih | kg | (42) |
| $m^{\text{H}_2\text{O}}$ | Mass of water | kg | (40) |
| $m^{\text{S}}$ | Mass of sea salt | kg | (40) |
| $m^{\text{V}}$ | Mass of water vapour | kg | (42) |

| Symbol | Quantity | SI Unit | Equation |
|---|---|---|---|
| $m^{\mathrm{W}}$ | Mass of liquid water | kg | (42) |
| $N$ | Number of particles | 1 | (41) |
| $N^{\mathrm{A}}$ | Number of dry-air particles | 1 | (41) |
| $N^{\mathrm{S}}$ | Number of sea salt particles | 1 | (41) |
| $N^{\mathrm{H_2O}}$ | Number of water molecules | 1 | (41) |
| $n_i^{\mathrm{A}}$ | Coefficients oft he Helmholtz function of dry air | 1 | (A16) |
| $n_i^{\mathrm{V}}$ | Coefficients oft he Helmholtz function of water vapour | 1 | (A17) |
| $p$ | Pressure | Pa | (13) |
| $p_{\mathrm{SO}}$ | Standard ocean surface pressure, $p_{\mathrm{SO}} = 101{,}325$ Pa | Pa | (65) |
| $p_{\mathrm{t}}$ | Triple-point pressure of water, $p_{\mathrm{t}} = 611.657$ Pa | Pa | (A38) |
| $p^*$ | Scaling pressure, $p^* = 10^8$ Pa | Pa | (A2) |
| $Q_L$ | Latent heat flux density | W m$^{-2}$ | (1) |
| $Q^{\mathrm{SA}}$ | Statistical configuration integral of sea air | 1 | (40) |
| $Q_V$ | Liquid water evaporation velocity | m s$^{-1}$ | (1) |
| $q$ | Specific (or absolute) humidity | kg kg$^{-1}$ | (1) |
| $\boldsymbol{q}$ | Vector of positions and orientations | | (41) |
| $q^{\mathrm{sat}}$ | Saturation specific humidity | kg kg$^{-1}$ | (32) |
| $q_{\mathrm{eq}}$ | Equilibrium specific humidity | kg kg$^{-1}$ | (1) |
| $R$ | Molar gas constant, $R = 8.314\,462\,618\,153\,24$ J mol$^{-1}$ K$^{-1}$ | J (mol K)$^{-1}$ | (13) |
| $R^{10}$ | 2010 molar gas constant, $R^{10} = 8.314\,472$ J mol$^{-1}$ K$^{-1}$ | J (mol K)$^{-1}$ | (A13) |
| $R^{95}$ | Molar gas constant of [109], $R^{95} = R_{\mathrm{W}}^{95} \times M_{\mathrm{W}}$ | J (mol K)$^{-1}$ | (A14) |
| Re | Real part of a complex number | | (A37) |
| $r_{jk}$ | Coefficients of the specific Gibbs energy of ice Ih | J (kg K)$^{-1}$ | (A37) |
| $R^{\mathrm{L}}$ | Molar gas constant of [117], $R^{\mathrm{L}} = 8.314\,51$ J mol$^{-1}$ K$^{-1}$ | J (mol K)$^{-1}$ | (A13) |
| $R_{\mathrm{W}}$ | Specific gas constant of water, $R_{\mathrm{W}} = R / M_{\mathrm{W}}$ | J (kg K)$^{-1}$ | (13) |
| $R_{\mathrm{W}}^{95}$ | Specific gas constant of water [109], $R_{\mathrm{W}}^{95} = 461.518\,05$ J kg$^{-1}$K$^{-1}$ | J (kg K)$^{-1}$ | (A14) |
| $S$ | Specific (or absolute) salinity | kg kg$^{-1}$ | (1) |
| $S^*$ | Scaling salinity, $S^* = 40$ psu $= 0.040\,188\,617$ | kg kg$^{-1}$ | (A7) |
| $T$ | (Absolute) temperature, ITS-90 | K | (1) |
| $t$ | Celsius temperature, $t = T - 273.15$ K | °C | |
| $t_{\mathrm{dp}}$ | Dew-point Celsius temperature | °C | (71) |
| $T_{\mathrm{AV}}$ | Temperature of humid air | K | (10) |
| $T_{\mathrm{dp}}$ | Dew-point temperature | K | (66) |
| $T_{\mathrm{fp}}$ | Frost-point temperature | K | (67) |
| $t_k$ | Coefficients of the specific Gibbs energy of ice Ih | 1 | (A37) |

| Symbol | Quantity | SI Unit | Equation |
|---|---|---|---|
| $T_{\mathrm{mp}}$ | Melting-point temperature | K | (68) |
| $T_{\mathrm{ref}}$ | Reference temperature | K | |
| $T_{\mathrm{SO}}$ | Standard ocean temperature, $T_{\mathrm{SO}} = 273.15$ K | K | (A2) |
| $T_{\mathrm{SW}}$ | Temperature of seawater | K | (10) |
| $T_{\mathrm{t}}$ | Triple-point temperature of water, $T_{\mathrm{t}} = 273.16$ K | K | (A37) |
| $T^*$ | Scaling temperature, $T^* = 40$ K | K | (A2) |
| $U$ | $N$-particle interaction potential | J | |
| $u$ | Wind speed | m s$^{-1}$ | (1) |
| $V$ | Volume | m$^3$ | (40) |
| $v^{\mathrm{W}}$ | Specific volume of liquid water | m$^3$ kg$^{-1}$ | (A4) |
| $X_k$ | Onsager force | various | (6) |
| $X_{\mathrm{Q}}$ | Onsager force of sensible heat flux | (K m)$^{-1}$ | (9) |
| $X_{\mathrm{W}}$ | Onsager force of water diffusion flux | J (kg K m)$^{-1}$ | (8) |
| $x_{\mathrm{V}}$ | Mole fraction of water vapour in humid air | mol mol$^{-1}$ | (13) |
| $x_{\mathrm{V}}^{\mathrm{sat}}$ | Saturation mole fraction of water vapour in humid air | mol mol$^{-1}$ | (21) |
| $x_{\mathrm{W}}$ | Mole fraction of water in seawater | mol mol$^{-1}$ | (14) |
| $z$ | Vertical coordinate | m | (8) |
| $\varepsilon_{\mathrm{V}}$ | Numerically negligible historical deviation | 1 | (A30) |
| $\varepsilon_f$ | Numerically negligible historical deviation | 1 | (A31) |
| $\eta_0$ | TEOS-10 specific residual entropy of ice Ih | J (kg K)$^{-1}$ | (A37) |
| $\eta^{\mathrm{SW}}$ | Specific entropy of seawater | J (kg K)$^{-1}$ | (A10) |
| $\eta^{\mathrm{W}}$ | Specific entropy of liquid water | J (kg K)$^{-1}$ | (A5) |
| $\lambda$ | 2-box water–air interface thickness | m | (10) |
| $\mu_{\mathrm{W}}$ | Specific chemical potential of water | J kg$^{-1}$ | (8) |
| $\mu_{\mathrm{W}}^{\mathrm{id}}$ | Ideal gas part of the chemical potential of water | J kg$^{-1}$ | (13) |
| $\mu_{\mathrm{W}}^{\mathrm{AV}}$ | Specific chemical potential of water in humid air | J kg$^{-1}$ | (10) |
| $\mu^{\mathrm{Ih}}$ | Specific chemical potential of ice Ih | J kg$^{-1}$ | (50) |
| $\mu_{\mathrm{W}}^{\mathrm{SW}}$ | Specific chemical potential of water in seawater | J kg$^{-1}$ | (10) |
| $\mu_{\mathrm{W}}^{\mathrm{W}}$ | Specific chemical potential of liquid water | J kg$^{-1}$ | (A5) |
| $\pi$ | Number Pi, $\pi = 2\arcsin(1) = 3.14\ldots$ | 1 | (41) |
| $\rho$ | Mass density of humid air | kg m$^{-3}$ | (A12) |
| $\rho^{\mathrm{A}}$ | Partial mass density of dry air | kg m$^{-3}$ | (A13) |
| $\rho^{\mathrm{AV}}$ | Mass density of humid air | kg m$^{-3}$ | (1) |
| $\rho^{\mathrm{Ih}}$ | Mass density of ice Ih | kg m$^{-3}$ | (42) |
| $\rho^{\mathrm{SW}}$ | Mass density of seawater | kg m$^{-3}$ | (42) |
| $\rho^{\mathrm{V}}$ | Partial mass density of water vapour | kg m$^{-3}$ | (A14) |
| $\rho^{\mathrm{W}}$ | Mass density of liquid water | kg m$^{-3}$ | (1) |
| $\rho^*$ | Scaling mass density, $\rho^* = 1$ kg m$^{-3}$ | kg m$^{-3}$ | (A13) |

| Symbol | Quantity | SI Unit | Equation |
|---|---|---|---|
| $\rho_A^*$ | Scaling air density, $\rho_A^* = \rho_{AA}^* \times M_A$ | $\mathrm{kg\ m^{-3}}$ | (A16) |
| $\rho_{AA}^*$ | Scaling molar density, $\rho_{AA}^* = 10{,}447.7\ \mathrm{mol\ m^{-3}}$ | $\mathrm{kg\ m^{-3}}$ | (A18) |
| $\rho_{AW}^*$ | Scaling molar density, $\rho_{AW}^* = 10^6\ \mathrm{mol\ m^{-3}}$ | $\mathrm{mol\ m^{-3}}$ | (A20) |
| $\rho_{WW}^*$ | Scaling molar density, $\rho_{WW}^* = \left(322\ \mathrm{kg\ m^{-3}}\right)/M_W$ | $\mathrm{mol\ m^{-3}}$ | (A19) |
| $\sigma$ | Entropy production density | $\mathrm{W\ K^{-1}\ m^{-3}}$ | (7) |
| $\tau$ | Temperature variable, $\tau = (132.6312\ \mathrm{K})/T$ | 1 | (A16),(A18) |
| $\tau$ | Temperature variable, $\tau = (647.096\ \mathrm{K})/T$ | 1 | (A17),(A19) |
| $\tau$ | Temperature variable, $\tau = T/(100\ \mathrm{K})$ | 1 | (A20) |
| $\psi_f$ | Relative fugacity | $\mathrm{Pa\ Pa^{-1}}$ | (22) |
| $\psi_q$ | Relative humidity (climatological definition) | $\mathrm{kg\ kg^{-1}}$ | (3) |
| $\psi_x$ | Relative humidity (metrological definition) | $\mathrm{Pa\ Pa^{-1}}$ | |
| $\Omega_{kl}$ | Onsager coefficient | various | (6) |
| $\Omega_{WW}$ | Onsager coefficient of irreversible evaporation | $\mathrm{J\ K\ (m\ s)^{-1}}$ | (12) |

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
