# Peer review of "Thermodynamics of Evaporation from the Ocean Surface"

_atmosphere, doi:10.3390/atmos14030560_

Round 1
Reviewer 1 Report
My comments are described in the attached pdf file.

Reviewer 2 Report
Review of “Thermodynamics of Evaporation from the Ocean Surface”
by Rainer Feistel and Olaf Hellmuth
This is a very important paper and certainly be published. In previous papers by the three global experts in this field, namely Feistel, Hellmuth and Lovell-Smith, they have shown that the rate of evaporation for a liquid surface into the humid atmosphere depends on the relative fugacity, not on relative humidity (kg/kg) or the partial pressure of water vapour in the humid air. Now, importantly, they examine the importance of this finding on the coupled air-sea-ice models that are used by the IPCC to evaluate future climate scenarios.
The present paper finds that the present practice in these models, upon which governments are making their decisions regarding the rate of run-down of the burning of fossil fuels, is flawed. These models use one assumption to calculate the air-sea flux of water, but this assumption responds INCORRECTLY to the warming of the sea surface temperature.
The publication of this paper should send shock waves through the coupled modelling community and cause them to immediately re-code the way that the rate of evaporation is calculated. This is an extremely important piece of research by the global experts in this field. The TEOS-10 standard for seawater, ice and humid air has been the approved global standard since 1st January 2010, and this paper shows another important consequence of the well-founded thermodynamic principles that are available with TEOS-10.
Many things in climate science and in coupled modelling are hard to get as accurate as we would like, but we should not willfully continue practices and parametrizations that fly in the face of fundamental physical and chemical laws. The present manuscript points out that the community has been doing this when estimating the evaporation rate in coupled models. This paper derives very easy ways in which coupled models can now properly incorporate the consequences of the fundamental and unavoidable thermodynamic principles.
To put this another way, we might say that to date atmospheric scientists have played fast and loose in their definition of “relative humidity”; there are three definitions and only one of them is the appropriate definition for estimating the rate of evaporation. The present paper shows an important consequence where this wooly thinking about “relative humidity” has come home to roost. Such inaccurate thinking cannot persist if we are to trust future scenarios of climate change from coupled models. Hence, this is a very important paper, and it’s conclusions should be adopted by climate modellers as soon as possible.
Here are some detailed comments on the manuscript.
Line 127, in Eqn. (2) a value of salinity seems to have been assumed.
Line 428 (and previously) the variable composition of sea salt has been overlooked in this paper, and yet this was a very important aspect of TEOS-10. This needs to be discussed in this paper.
Line 454, Eqn. (59) The derivation of this latent heat of melting is very succinct in the present paper. I believe that the first derivation of this accurate expression was reference [19] which was a much more detailed derivation. Please give the reader some more help in understanding the derivation.
Reviewer 3 Report
The manuscript contains interesting analysis of irreversible thermodynamics of water evaporation from the ocean surface and provides a detailed information about thermodynamic properties of humid air, seawater, and ice, based on the TEOS-10 standard.
I have some suggestions for additional considerations and optional completing the text.
1) In Eq. (10) you define gradient driving the mass transfer, which is based on the difference of chemical potentials. This leads to Eq. (19), which gives the mass flux in terms of a ratio of fugacities. As you write in Eq. (27), for practical application the ratio of fugacities can be replaced by a ratio of molar fractions, giving J=-D_f ln( x_v/x_v^sat). However, as far as I understand the kinetic theory of gases, diffusion flux is perfectly proportional to the negative of gradient of molar fraction, or to a difference of molar fractions at two points in the molar fraction profile, i.e., proportional to (x_v^sat-x_v). The difference can be obtained by linearizing the logarithm, but I think that the difference is physical (in gases at atmospheric pressure) rather than the logarithm. The situation is not qualitatively changed in turbulent diffusion, only the rate coefficient D_f depends not only on the diffusion coefficient, but also on the wind speed and air viscosity. Please re-think this.
2) At line 191 your write "Further, \lambda is the thickness of a fictious membrane separating the boxes...". While it is possible to look at it in a mathematical abstraction, I think it should also be said that this quantity physically depends on the parameters of turbulent diffusion in the atmospheric boundary layer. It would be interesting to consider the mass transfer in terms of dimensionless Sherwood, Schmidt, and Reynolds numbers, as it is usual in mass transfer calculations.
3) Around Eq. (24) you argue that D_f is constant with respect to global warming trend, rather that D_e introduced later. In fact, this argument is based on an observation (really correct?) that climatological evaporation rate can be considered as a constant, although the global temperature changes. In more physical terms this assumes that evaporation rate is independent of temperature. This seems to violate common observations, obviously evaporation rate rapidly increases with temperature.
4) The virial coefficient of water vapor B^ww is given by Eq. (B.8). I think that more accurate and much simpler is formulation by Harvey and Lemmon, J. Phys. Chem. Ref. Data. 33 (2004) 369. This formulation should at least be mentioned. For water-air, recent data and equation is by Hellmann, J. Chem. Eng. Data 65 (2020) 4130−4141.
Reviewer 4 Report
The paper presents the state of the art for the thermodynamics of evaporation from the ocean surface. The paper is of high quality and should be published as is.
The structure of the paper is well organized. The authors introduce the topic and gave some important numbers from literature to the reader. In the second paragraph the Dalton equation and its application to evaporation at the ocean surface are discussed. The third paragraph of the paper introduces more accurate state-of-the-art models for the thermodynamics of seawater, humid air, and ice as well as applied approximations. These models are adopted as standards of the leading organizations and therefor it is recommendable to apply these models for the topic discussed in this paper. In the fourth paragraph an equation of state is introduces followed by its application for latent heat which is the key property for calculating evaporation processes. Finally, the fifth paragraph deals with the approximation of the relative fugacity to Daltons law. The conclusion summarizes the findings of this paper.
The derivations of the formulas given in the paper are clear and strictly provided. The symbols used in the paper are appropriate and the indices make clear to which phase the property is belonging to. In addition, the symbols are introduces when they are used for the first time. Maybe a list of symbols could help to improve the understanding by readers who are not familiar to the symbols and indices.
The appendix provides all numerical details to implement the equations given in the paper and the same notation was used as in the paper.
Concluding this is a very good presented paper with a highly important content to the community dealing with climatic modelling. Its publication is highly recommended.
Round 2
Reviewer 1 Report
I am satisfied with the answers to my review. I believe that persons reading the paper in the journal "Atmosphere" would benefit if the clear and compact author's answers to my questions 1 and 2 were introduced in the introduction and summary.
